# Synergistic effect of inhibiting CHK2 and DNA replication on cancer cell growth

**Flavie Coquel[1,2‡], Sing-Zong Ho[3], Keng-Chang Tsai[4,5], Chun-Yen Yang[1], Antoine Aze[1,6], Julie Devin[1,7], Ting-Hsiang Chang[3], Marie Kong-Hap[8], Audrey Bioteau[1,2], Jerome Moreaux[1,7,9,10], Domenico Maiorano[1,6], Philippe Pourquier[8], Wen-Chin Yang[3,11,12]\*, Yea-Lih Lin[2]\*†, Philippe Pasero[1,2]\*†**

[1]Institut de Génétique Humaine, Univ. de Montpellier, CNRS, Montpellier, France; [2]'Maintenance of Genome Integrity during DNA replication' laboratory, équipe labélisée Ligue contre le Cancer, Montpellier, France; [3]Agricultural Biotechnology Research Center, Academia Sinica, Taipei, Taiwan; [4]Ph.D. Program in Medical Biotechnology, College of Medical Science and Technology, Taipei Medical University, Taipei, Taiwan; [5]National Research Institute of Chinese Medicine, Ministry of Health and Welfare, Taipei, Taiwan; [6]'Genome Surveillance and Stability' Laboratory, IGH, Univ. de Montpellier, CNRS, Montpellier, France; [7]'Normal and Malignant B cells' laboratory', IGH, Univ. de Montpellier, CNRS, Montpellier, France; [8]IRCM, Institut de Recherche en Cancérologie de Montpellier, INSERM U1194, Université de Montpellier, Institut régional du Cancer de Montpellier, Montpellier, France; [9]Institut Universitaire de France, Paris, France; [10]Department of Biological Hematology, CHU Montpellier, Montpellier, France; [11]Graduate Institute of Integrated Medicine, China Medical University, Taichung, Taiwan; [12]Department of Life Sciences, National Chung-Hsing University, Taichung, Taiwan

**\*For correspondence:**
wcyang@gate.sinica.edu.tw (W-CY);
yea-lih.lin@igh.cnrs.fr (Y-LL);
philippe.pasero@igh.cnrs.fr (PP)

†These authors contributed equally to this work

**Present address:** ‡Incyte Biosciences, Yverdon-les-Bains, Switzerland

**Competing interest:** The authors declare that no competing interests exist.

## eLife Assessment

This study presents **important** findings on the activity of two compounds, BKC and IBC, isolated from *Psoralea corylifolia*, which act synergistically to inhibit cancer cell proliferation. Using a spectrum of methods, the authors characterized the mechanisms of action of both drugs, providing **convincing** evidence that BKC targets DNA polymerases and IBC selectively inhibits CHK2. The study opens the possibility of improving the effectiveness of the combination of BKC and other damaging agents with IBC in cancer treatment.
[Editors' note: this paper was reviewed by Review Commons.]

**Abstract** Cancer cells display high levels of oncogene-induced replication stress (RS) and rely on DNA damage checkpoint for viability. This feature is exploited by cancer therapies to either increase RS to unbearable levels or inhibit checkpoint kinases involved in the DNA damage response. Thus far, treatments that combine these two strategies have shown promise but also have severe adverse effects. To identify novel, better-tolerated anticancer combinations, we screened a collection of plant extracts and found two natural compounds from the plant, *Psoralea corylifolia*, that synergistically inhibit cancer cell proliferation. Bakuchiol inhibited DNA replication and activated the checkpoint kinase CHK1 by targeting DNA polymerases. Isobavachalcone interfered with DNA double-strand break repair by inhibiting the checkpoint kinase CHK2 and DNA end resection. The combination of bakuchiol and isobavachalcone synergistically inhibited cancer cell proliferation in vitro. Importantly, it also prevented tumor development in xenografted NOD/SCID mice. The synergistic effect of inhibiting DNA replication and CHK2 signaling identifies a

vulnerability of cancer cells that might be exploited by using clinically approved inhibitors in novel combination therapies.

## Introduction

Genome integrity is particularly at-risk during S phase of the cell cycle when thousands of replication forks travel at high speed along the chromosomes to duplicate the DNA. Replication initiates at specific sites called origins, which are sequentially activated throughout the length of the S phase (*Fragkos et al., 2015*). Faithful DNA replication depends on the coordinated action of the many enzymes that make up the replisome (*Attali et al., 2021*). During this process, the parental DNA strands are separated by the CMG helicase, consisting of CDC45, the MCM2-7 hexamer, and the GINS complex. DNA is synthesized on the leading strand by DNA polymerase (Pol) ε and on the lagging strand by Pol α-primase and Pol δ (*Lujan et al., 2016*).

Replication forks stall when they encounter obstacles such as DNA lesions, highly transcribed genes, or tightly bound protein complexes, causing what is commonly referred to as replication stress (RS) (*Lin and Pasero, 2021*; *Macheret and Halazonetis, 2015*; *Zeman and Cimprich, 2014*). RS is associated with an excess of single-stranded DNA (ssDNA), resulting from the uncoupling of DNA polymerase and helicase activities (*Pasero and Vindigni, 2017*). Stalled or collapsed forks can also give rise to DNA double-strand breaks (DSBs). A signal transduction pathway called the intra-S checkpoint detects these stalled forks and associated DSBs (*Zeman and Cimprich, 2014*). Upon fork arrest, ssDNA coated by the ssDNA-binding heterotrimer RPA recruits the Ser/Thr protein kinase ATR (*Zou and Elledge, 2003*). Activation of ATR by TopBP1 initiates a signaling cascade involving phosphorylation and activation of the intra-S checkpoint kinases CHK1 and WEE1, which coordinate a variety of repair mechanisms to prevent fork collapse, resume DNA synthesis, and delay entry into mitosis (*Pasero and Vindigni, 2017*; *Saldivar et al., 2017*; *Stracker et al., 2008*). The presence of DSBs is also signaled by two other protein kinases, ATM and DNA-PK, which activate the checkpoint kinase CHK2 and result in cell cycle arrest or cell death (*Blackford and Jackson, 2017*). ATR, ATM, and DNA-PK phosphorylate the histone variant H2AX on S139 (γ-H2AX), which is a reliable biomarker of RS and DSBs (*Bonner et al., 2008*).

During the S and $G_2$ phases of the cell cycle, DSBs are preferentially repaired by homologous recombination (HR), which uses the sister chromatid as repair template (*Moynahan and Jasin, 2010*). In $G_1$ phase or in HR-deficient cells, DSBs are repaired by more error-prone pathways such as non-homologous end joining and single-strand annealing (*Chang et al., 2017*). HR-mediated DSB repair is initiated with the resection of DNA ends to generate 3'-protruding extremities that are coated by the RAD51 recombinase to form a RAD51 filament involved in homology search (*Cejka and Symington, 2021*; *Chakraborty et al., 2023*). RAD51 loading depends on BRCA1 and BRCA2, which also play a role at stalled replication forks to prevent the hyper-resection of nascent DNA (*Tye et al., 2021*).

In precancerous lesions, deregulated oncogenic pathways perturb the proper execution of the DNA replication program, leading to increased fork collapse and chromosome breaks (*Macheret and Halazonetis, 2015*). This oncogene-induced RS, on the one hand, promotes cancer development by increasing genomic instability and promoting the loss of p53 *Halazonetis et al., 2008*; on the other hand, it is a burden for cancer cells, which must deal with chronic replication defects and may become dependent on checkpoint function for their survival. This burden can be exploited for cancer treatment by the use of genotoxic drugs that further increase RS and/or by inhibiting the ATR pathway (*Lecona and Fernandez-Capetillo, 2018*; *Ubhi and Brown, 2019*; *Zhu et al., 2020*). Cancer cells eventually adapt to oncogene-induced RS, however, by overexpressing downstream components of the ATR pathway, such as Claspin, Timeless, and CHK1, which correlates with poor prognosis in breast, lung, and colon carcinomas (*Bianco et al., 2019*). Thus, genotoxic agents used to increase RS in cancer cells are generally effective as first-line treatments to reduce tumor mass, but patients often relapse as the cancer cells become resistant to treatment.

Since cancer cells depend on an effective RS response for their survival, considerable effort has been made to develop small-molecule inhibitors of the intra-S checkpoint kinases ATR, CHK1, and WEE1 (*Dobbelstein and Sørensen, 2015*; *Lecona and Fernandez-Capetillo, 2018*; *Ubhi and Brown, 2019*). In principle, inhibiting these kinases should selectively kill cancer cells that have elevated levels of RS while sparing healthy cells. Several inhibitors have now entered clinical trials, with mixed results

(*Bradbury et al., 2020*; *Gorecki et al., 2021*). ATR inhibitors are usually well tolerated when used in monotherapy, but they have limited efficacy because low ATR activity can be compensated by ATM. ATR inhibitors are more effective when combined with subtherapeutic doses of chemotherapeutic agents (e.g., gemcitabine), but at the expense of serious adverse effects, such as myelosuppression (*Dobbelstein and Sørensen, 2015*; *Gorecki et al., 2021*; *Nazareth et al., 2019*). CHK1 inhibitors are toxic, especially when used in combinatorial therapies (*Neizer-Ashun and Bhattacharya, 2021*). WEE1 inhibitors are better tolerated when used in monotherapy and combination regimens, but clinical trials have shown only modest benefits so far (*Gorecki et al., 2021*). Thus, although some of these inhibitors are promising, they are still far from ready to be used in the clinic. Further efforts are still needed to identify new small-molecule inhibitors, or combinations of inhibitors, that target the RS response in cancer cells but are not excessively toxic to normal cells.

To identify novel small molecules that target the RS response in cancer cells, we screened crude extracts of plants used in traditional Chinese herbal medicine for their ability to kill cancer cells selectively by inducing RS. An extract from the plant *Psoralea corylifolia* showed the most promising anticancer activity. We show that bakuchiol (BKC) and isobavachalcone (IBC), two compounds isolated from this extract, act synergistically to prevent the proliferation of cancer cells, by inhibiting DNA synthesis and impeding the resection of DNA ends at DSBs, respectively. Together, these compounds reduce tumor growth and improve survival in a xenograft mouse model and IBC alone potentiates the anticancer effect of chemotherapeutic agents in diffuse large B-cell lymphoma (DLBCL) cells. The synergistic effect of an inhibitor of DNA synthesis and an inhibitor of DNA end resection identifies a novel vulnerability of cancer cells that might be exploited by using clinically approved inhibitors of these mechanisms in novel combination therapies.

## Results
### IBC and BKC synergistically inhibit proliferation of cancer cell lines

To identify novel combinations of small-molecule inhibitors that target DNA replication in cancer cells, we screened a selection of crude extracts of Chinese herbal medicines for their ability to differentially impede cell growth and induce γ-H2AX foci in MCF-7 human breast cancer cells relative to non-cancerous BJ hTERT-immortalized human fibroblasts. In an extract prepared from *P. corylifolia*, we identified two compounds, IBC (MW: 324.4; *Kuete and Sandjo, 2012*) and BKC (MW: 256.4; *Nizam et al., 2023*; *Figure 1A*), which inhibited the proliferation of MCF-7 cells and A549 human lung cancer cells more than they did the proliferation of BJ fibroblasts and non-transformed epithelial cells MCF10A and RPE-1 cells (*Figure 1B*, *Figure 1—figure supplement 1A and B*). BKC is a bioactive meroterpene that possesses a variety of pharmacological activities (*Xin et al., 2019*). It was shown to inhibit the proliferation of many cancer cell lines (*Li et al., 2016*) presumably through the inhibition of DNA replication (*Sun et al., 1998*). IBC is a natural chalcone that also exhibits potential anticancer activities (*Kuete et al., 2015*; *Ren et al., 2024*; *Wu et al., 2022*). However, the combined anticancer effect of BKC and IBC has never be addressed and their mechanisms of action and molecular targets have remained unknown.

Here, we have used a panel of eight tumor cell lines including two breast (MCF-7 and SUM159), one lung (HCC827), two prostate (PC3 and DU145), one lymphoma (U937), one colon (HCT116), and one ovarian (Ovcar8) cancer cell lines to investigate the antitumoral effect of IBC and BKC. We first determined the $IC_{50}$ of each compound in each cell line (*Figure 1—figure supplement 1C*) and found that IBC was more potent than BKC in inhibiting the proliferation of all cell lines. We then evaluated the effect of combined use of IBC and BKC on the viability of the eight cell lines by using a concentration matrix approach and a quantitative colorimetric cytotoxicity assay (*Figure 1—figure supplement 1D*) from which we calculated the synergistic and antagonistic effects of the two compounds (*Figure 1C*), as previously described (*Tosi et al., 2018*). In all cell lines tested, we observed a synergistic effect of IBC and BKC over a narrow range of concentrations (3–10 µM IBC and 10–30 µM BKC) and an additive effect of the two drugs over a wider concentration range. This ratio of 3:10 µM IBC:BKC that has a synergistic effect in vitro corresponds to the ratio of concentrations of the two compounds in the fruits of *P. corylifolia*. We conclude from this data that IBC and BKC inhibit cell growth in a synergistic manner and that this effect is more pronounced on cancer cells than on non-cancer cells.

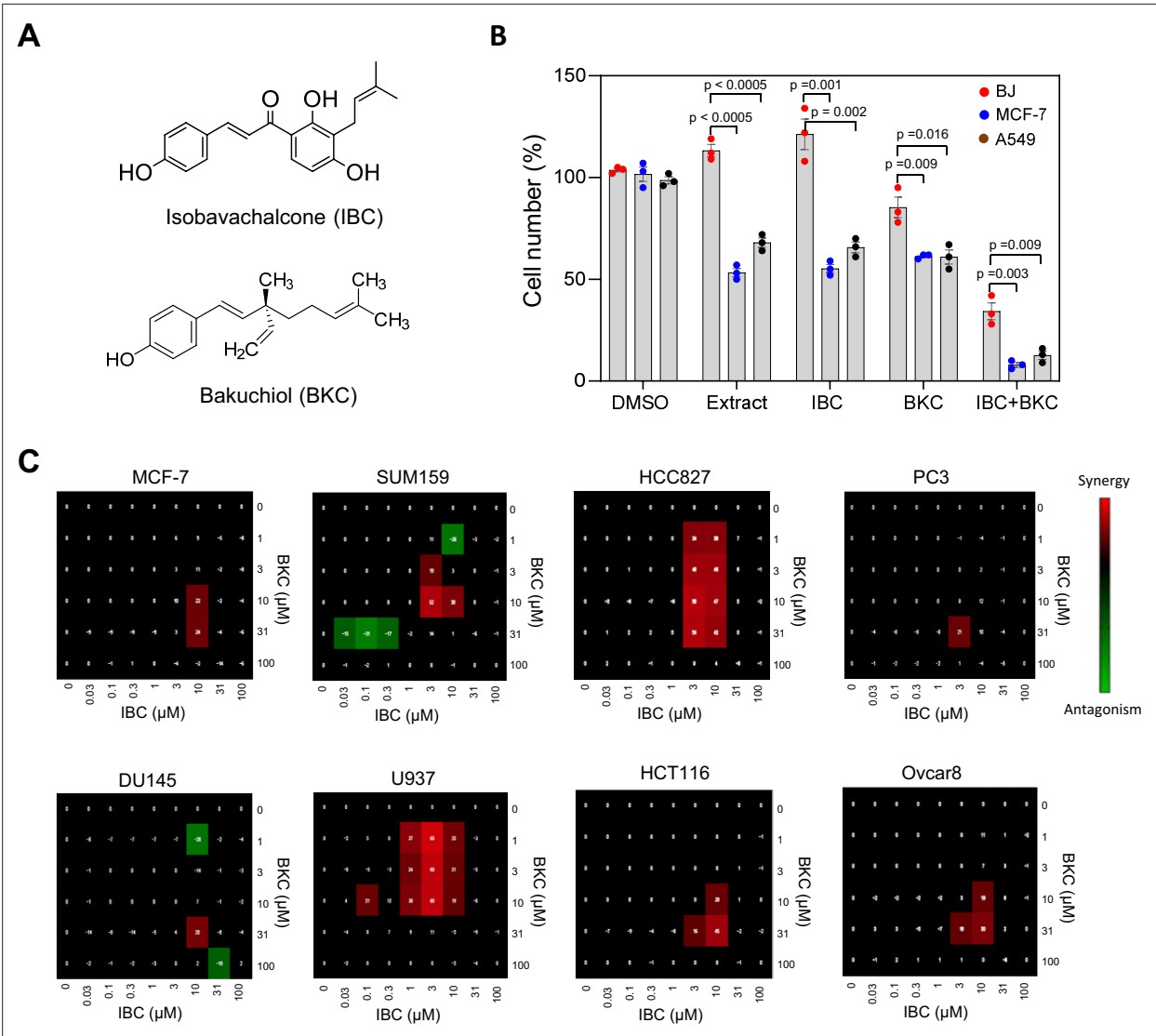

**Figure 1.** Isobavachalcone (IBC) and bakuchiol (BKC) synergistically inhibit proliferation of cancer cell lines. (**A**) Chemical structures of IBC and BKC. (**B**) BJ, MCF-7, and A549 cells were treated with DMSO, 25 µg/ml crude extract (PR7), 15 µM IBC, 40 µM BKC, or the combination 15 µM IBC and 40 µM BKC for 72 hr. These concentrations were used throughout the study. Cell number was quantified by using the WST-1 assay. Data are means ± SD of three independent experiments. The p-values were calculated using two-tailed unpaired *t*-test. (**C**) Concentration matrix analyses of a panel of eight cancer cell lines treated with IBC and BKC at the indicated doses for 72 hr. Cell viability was measured by using the sulforhodamine B colorimetric assay. Antagonist combinations (green), synergistic combinations (red), and additive effects (black) were calculated. A representative analysis of three independent experiments is shown.

The online version of this article includes the following figure supplement(s) for figure 1:

**Figure supplement 1.** Isobavachalcone (IBC) and bakuchiol (BKC) synergistically inhibit proliferation of cancer cell lines.

## IBC and BKC induce replication stress

To study the potential of IBC and BKC to induce RS, we assayed formation of γ-H2AX foci in MCF-7 and BJ cells treated for 24 hr with DMSO or with the two compounds, either alone or in combination. Cells were labeled with the thymidine analogue 5-ethynyl-2′-deoxyuridine (EdU) to identify cells in S phase and γ-H2AX foci were detected by immunofluorescence microscopy; γ-H2AX levels in S phase cells was quantified as mean fluorescence intensity. IBC increased γ-H2AX signal in MCF-7 but not in BJ cells, whereas BKC increased it in both cell types (***Figure 2A and B***). Moreover, the combination of IBC and BKC further increased γ-H2AX fluorescence in MCF-7 but not in BJ cells (***Figure 2A and B***), which is consistent with their effect on cancer cell growth. BKC alone or in combination with IBC

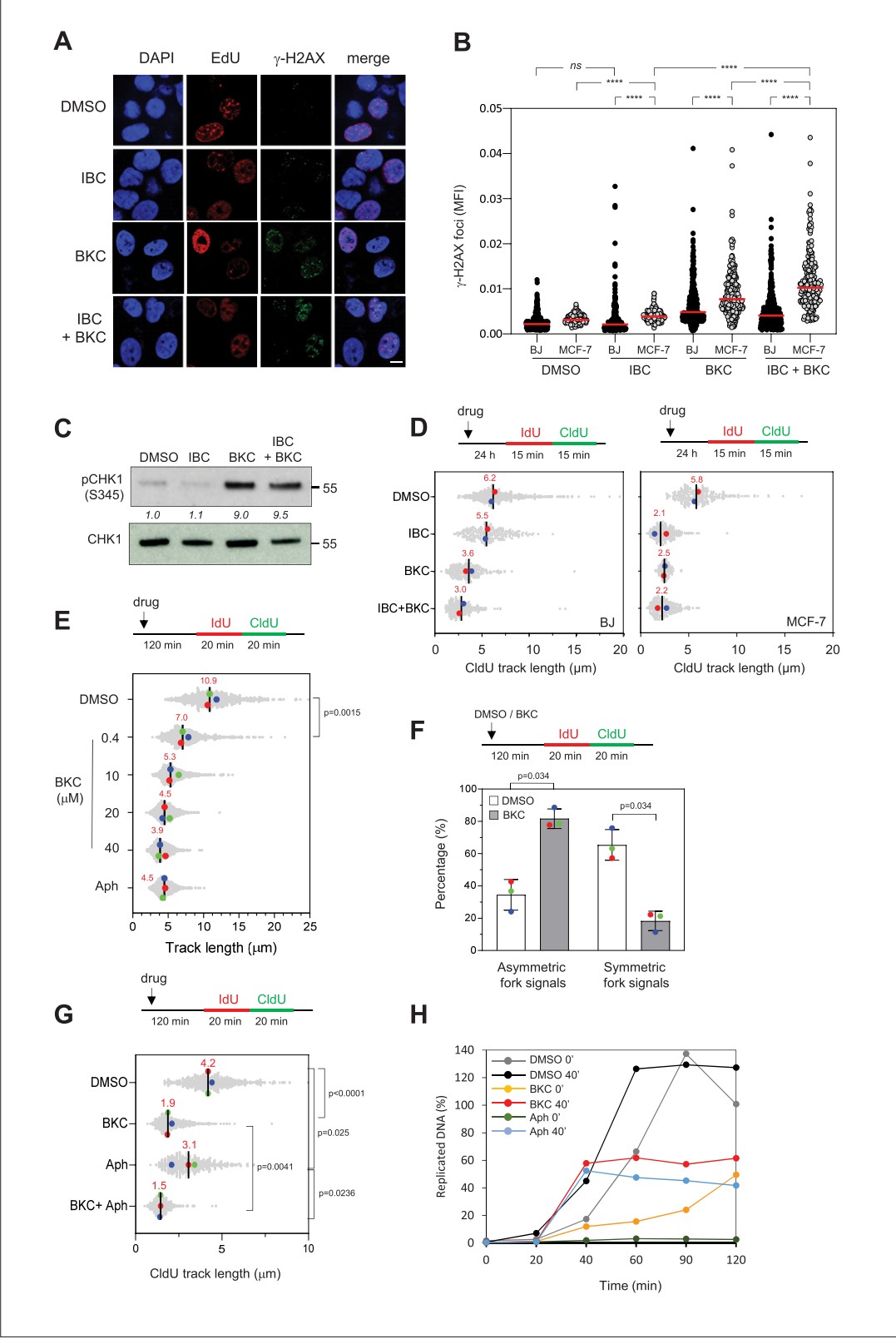

**Figure 2.** Isobavachalcone (IBC) and bakuchiol (BKC) induce replication stress and impede fork progression. (**A**) MCF-7 and BJ cells were treated with either 15 μM IBC, 40 μM BKC, or both (IBC + BKC) for 24 hr, then 10 μM EdU was added for 10 min and γ-H2AX foci in EdU-positive cells were detected by using Click chemistry and immunofluorescence microscopy. Representative immunofluorescence images of MCF-7 cells are shown.

*Figure 2 continued*

Bar: 5 µm. (**B**) Mean fluorescence intensity (MFI) of γ-H2AX foci was quantified using CellProfiler. One of three independent experiments is shown (n = 3). \*\*\*\*p<0.0001; *ns*, not significant, Mann–Whitney rank sum test. (**C**) MCF-7 cells were treated with IBC/BKC for 24 hr, as in (**A**), and CHK1 phosphorylated on S345 (pCHK1) was detected by western blotting. The ratio of pCHK1 to total CHK1, relative to the DMSO control, is indicated. A representative example of two independent experiments is shown. (**D**) BJ and MCF-7 cells were treated with IBC/ BKC for 24 hr, as in (**A**), then IdU and CldU were added sequentially each for 15 min. Replication fork progression was determined by measuring CldU track lengths in DNA fiber spreads. The median length of CldU tracks is indicated in red. At least 150 fibers were measured for each condition. Median of two independent experiments is indicated in red. (**E**) MCF-7 cells were treated with indicated concentrations of BKC or 1 µM aphidicolin (Aph) for 2 hr then IdU and CldU were added sequentially each for 20 min. Replication fork progression was determined as in (**D**). The length of IdU and CldU was measured. Median of three independent experiments is indicated in red. The p-values were determined using two-tailed unpaired *t*-test (n = 3). (**F**) MCF-7 cells were treated with DMSO or 20 µM BKC for 2 hr then IdU and CldU were added sequentially each for 20 min. Replication fork progression was determined as indicated in (**D**). The ratio of the CldU signal of two sister forks was calculated. At least 80 sister forks were measured in each biological replicate. The ratio of two sister forks between 0.8 and 1.2 was considered as symmetric forks. Mean ± SEM of three independent experiments are shown. The p-values were determined using two-tailed unpaired *t*-test. (**G**) MCF-7 cells were treated with 40 µM BKC, Aph (10 µM) or both (BKC + Aph) for 2 hr, prior to DNA fiber spreading assay. The p-values were determined using two-tailed unpaired *t*-test (n = 3). (**H**) *Xenopus* egg extracts were incubated with demembranated sperm nuclei and treated immediately (0 min) or after 40 min (40 min) with DMSO, BKC (100 µM), or Aph (60 µM). Samples were collected at the indicated time points after addition of the sperm nuclei. The percentage of replicated DNA was calculated as described in the 'Materials and methods'.

The online version of this article includes the following source data and figure supplement(s) for figure 2:

**Source data 1.** Original membranes corresponding to *Figure 2C* with labels.

**Source data 2.** Original membranes corresponding to *Figure 2C*.

**Figure supplement 1.** Bakuchiol (BKC) inhibits DNA replication and induces S-phase accumulation in the cell cycle.

---

also induced phosphorylation of CHK1 on S345 in MCF-7 cells, whereas IBC alone did not (*Figure 2C*, *Figure 2—figure supplement 1A*). These data suggest that BKC, but not IBC, can directly induce RS.

To address this possibility, we measured the effect of IBC and BKC on replication fork progression in MCF-7 and BJ cells by using a DNA fiber spreading assay. Briefly, the cells were exposed to one or both drugs for 24 hr and then labeled sequentially with the thymidine analogues 5-iodo-2′-deoxyuridine (IdU) and 5-chloro-2′-deoxyuridine (CldU) each for 15 min and the length of replicated tracks was measured along individual DNA fibers (*Figure 2D*). Analysis of CldU track length showed that BKC reduced fork speed by a factor of two relative to untreated cells in both cell lines, suggesting that it directly inhibits DNA synthesis. In contrast, IBC inhibited fork speed more in MCF-7 cells than in BJ cells, indicating that it may affect DNA synthesis through a mechanism different from that of BKC. When used in combination, the inhibitory effect of BKC and IBC on fork speed was further increased, consistent with our finding that these drugs have a synergistic effect on cell growth.

To evaluate the impact of IBC and BKC on the cell cycle, we exposed MCF-7 and BJ cells to these drugs for 24 hr, labeled the cells in S phase with EdU for 30 min and analyzed the distribution of cells in the various phases of the cell cycle by flow cytometry (*Figure 2—figure supplement 1B*). Treatment with BKC, but not IBC, resulted in a greater proportion of both cell types in S phase, which is consistent with our observation that BKC, but not IBC, induced CHK1 activation (*Figure 2C*, *Figure 2— figure supplement 1B*). Together, these data suggest that IBC and BKC induce RS through different mechanisms to prevent cancer cell proliferation.

## BKC inhibits DNA replication

Our data suggest that BKC might be a potent inhibitor of replication fork progression in vivo (*Figure 2D*) that acts directly on replicative DNA polymerases. Using DNA fiber assay, we found that BKC inhibited fork progression in a dose-dependent manner (*Figure 2E*). Furthermore, BKC treatment significantly increased sister fork asymmetry compared to DMSO-treated control cells (*Figure 2F*). We then compared its effect to that of aphidicolin, a well-characterized inhibitor of DNA polymerases a, d, and ε (*Cheng and Kuchta, 1993*). MCF-7 cells were treated with 40 µM BKC, 10 µM

aphidicolin, or both for 2 hr and replication fork progression was measured by DNA fiber spreading. Remarkably, BKC inhibited fork progression more effectively than aphidicolin at these concentrations and the combined effect of both compounds was similar to the effect of BKC alone (*Figure 2G*). BKC is a phenolic compound structurally related to resveratrol. Since resveratrol was shown to induce RS by inhibiting dNTP synthesis (*Benslimane et al., 2020*; *Fontecave et al., 1998*), we tested the possibility that BKC might also impede DNA replication by inhibiting dNTP synthesis. To this end, we tested its effect on a replication assay in *Xenopus* egg extracts, which contain high concentrations of dNTPs and do not depend on dNTP synthesis to sustain effective DNA replication. In this assay, demembranated sperm nuclei incubated in egg extracts decondense, assemble pre-replication complexes within 20 min and initiate synchronous DNA synthesis (*Méchali and Harland, 1982*). When added at the start of the assay (0 min), BKC (100 µM) exhibited a strong inhibitory effect on DNA replication, although not as profound as the effect of aphidicolin (60 µM). Moreover, when added after 40 min, long after the initiation of replication, BKC was as effective as aphidicolin (*Figure 2H*), suggesting that it inhibits elongation. Consistent with this possibility, BKC effectively inhibited replication of ssDNA, a process that relies entirely on priming and elongation of DNA chains by replicative DNA polymerases (*Figure 2—figure supplement 1C*).

We also performed in silico molecular docking to study the potential interaction of BKC with the catalytic subunits of DNA polymerases δ and ε. This analysis showed that BKC can occupy the deoxycytidine sites of both enzymes (*Figure 3A and B*). Using the cellular thermal sensitivity shift assay (CETSA) (*Martinez Molina et al., 2013*), we found that BKC interacts with the catalytic subunits of Polδ and Polε in MCF-7 cells to stabilize the thermal sensitivity of both proteins (*Figure 3C and D*). Importantly, BKC specifically stabilize the catalytic subunit POLD1 of Polδ, but not that of the accessory subunit POLD3 in *Xenopus* egg extracts (*Figure 3—figure supplement 1A and B*). Similarly, BKC did not alter the thermal sensitivity of *Xenopus* PCNA (*Figure 3—figure supplement 1C*). Together, these findings indicate that BKC strongly inhibits DNA replication in vivo and in vitro, most likely by directly inhibiting DNA polymerases.

## IBC inhibits CHK2

Previous studies found that IBC impedes cell proliferation by inhibiting AKT, a protein kinase involved in cell survival and in the transcriptional regulation of DNA replication-associated genes (*Jing et al., 2010*; *Spangle et al., 2016*). To determine whether AKT inhibition accounts for the effect of IBC on DNA replication in MCF-7 cells, we assayed the effect of IBC on the autophosphorylation of endogenous AKT at S473 and compared it to the effect of an allosteric AKT inhibitor, MK-2206. Whereas MK-2206 strongly inhibited AKT phosphorylation, IBC had no effect when used at the concentration that inhibited cell proliferation (30 µM), nor did BKC or a combination of both compounds (*Figure 4—figure supplement 1A*). Moreover, unlike MK-2206, IBC had little or no effect on expression of the cell cycle genes encoding E2F1, E2F2, and PCNA (*Figure 4—figure supplement 1B*). Also, unlike IBC, MK-2206 had no effect on replication fork speed (*Figure 4—figure supplement 1C*). We conclude that, at the concentrations used in this study, IBC inhibition of DNA replication cannot be explained by inhibition of AKT.

To identify candidate target(s) of IBC that are responsible for its inhibitory effect on cancer cell proliferation, we assayed the effects of a broad range of IBC concentrations on the activities of 43 cell cycle-related kinases in vitro and determined the $IC_{50}$ for each of them (*Figure 4A*, *Figure 4—figure supplement 1D and E*). The most sensitive kinase, CHK2, had an $IC_{50}$ for IBC of 3.5 µM (*Figure 4A*). Aurora-A/B and JNK3 were also sensitive to IBC, but at approximately fivefold higher concentrations, $IC_{50}$ of 11.2 µM for Aurora-A/B and 16.4 µM for JNK3. By contrast, CHK1 was not inhibited by IBC (*Figure 4A*). Consistent with our findings above, the $IC_{50}$ for AKT1/PKBα was 56.7 µM (*Figure 4A*, *Figure 4—figure supplement 1E and F*).

To validate the inhibitory effect of IBC on CHK2 in vivo, we analyzed the autophosphorylation of CHK2 on S516 induced by camptothecin (CPT), a DNA topoisomerase I inhibitor that induces RS and DSBs (*Pommier, 2006*). In MCF-7 cells, IBC inhibited the autophosphorylation of CHK2 induced by CPT by approximately 50% (*Figure 4B*, *Figure 4—figure supplement 1G*). Consistently, IBC also inhibited the activation of the downstream target of CHK2, BRCA1. We showed that treatment with IBC reduced the phosphorylation of chromatin-bound BRCA1 at residue S988 induced by CPT, as efficiently as the commercial CHK2 inhibitor BML-277 (*Figure 4C*, *Figure 4—figure supplement 1H*).

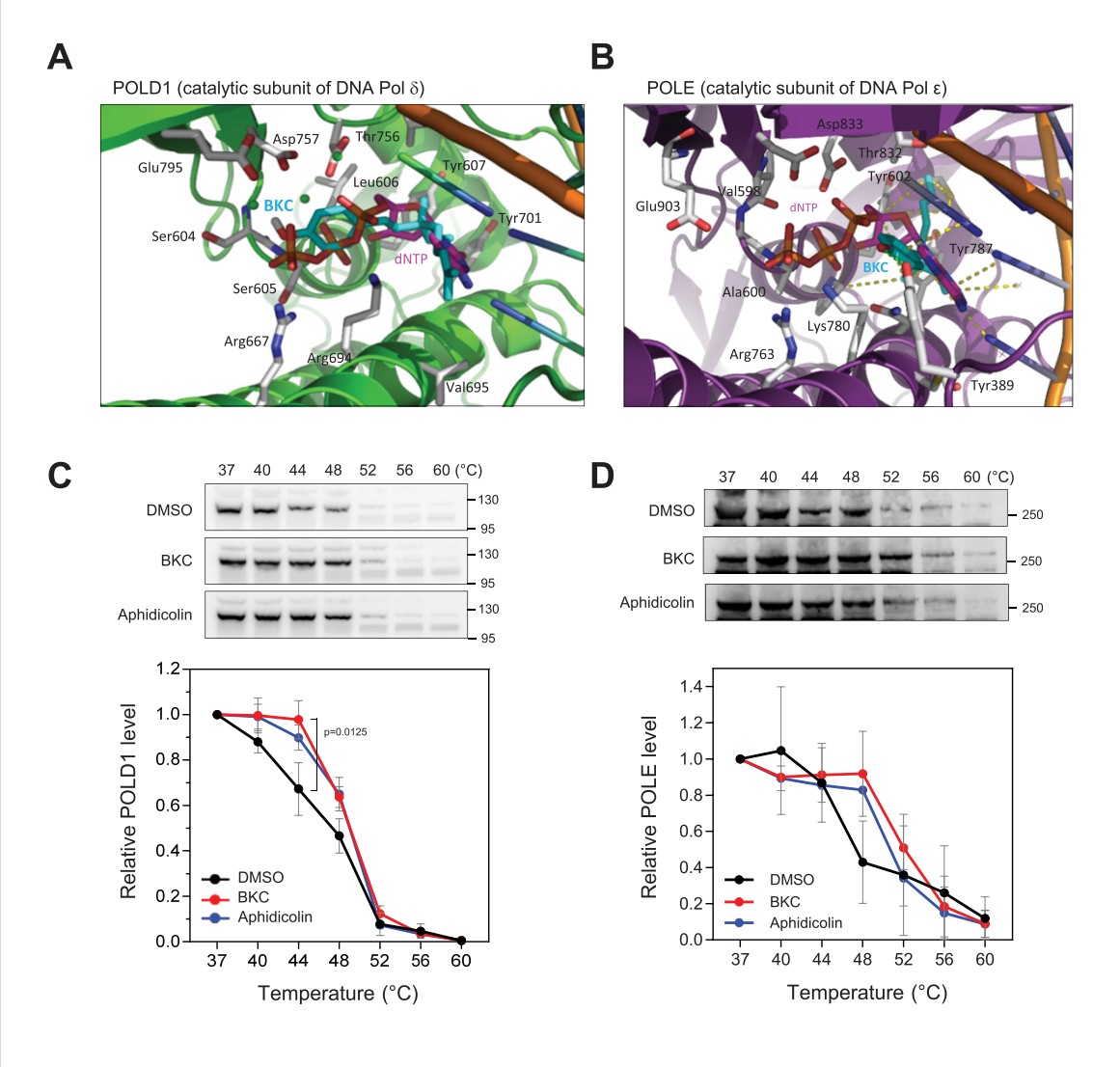

**Figure 3.** Bakuchiol (BKC) directly binds to DNA polymerases. (**A, B**) In silico molecular docking of bakuchiol in the predicted active site structures of human DNA Pol δ and ε, respectively. (**C, D**) MCF-7 cells were treated with DMSO, 40 µM BKC, or 10 µM aphidicolin for 2 hr prior to the cellular thermal sensitivity shift assay (CETSA) at indicated temperature, as described in the Materials and Methods. Levels of POLD1 (Pol δ; panel C)and POLE (Pol ε; **D**) catalytic subunits were detected by western blotting. Mean and SEM of three independent experiments are shown. The p-values were determined using two-tailed paired t-test.

The online version of this article includes the following source data and figure supplement(s) for figure 3:

**Source data 1.** Original membranes corresponding to *Figure 3C and D* with labels.

**Source data 2.** Original membranes corresponding to *Figure 3C and D*.

**Figure supplement 1.** Bakuchiol (BKC) does not interact with PCNA.

**Figure supplement 1—source data 1.** Original membranes corresponding to *Figure 3—figure supplement 1A–C* with labels.

**Figure supplement 1—source data 2.** Original membranes corresponding to *Figure 3—figure supplement 1A–C*.

By contrast, IBC did not affect the autophosphorylation of CHK1 on S296 induced by hydroxyurea (HU; *Figure 4D*, *Figure 4—figure supplement 1I*). Using in silico molecular docking, we found that IBC binds to the active site of CHK2 by means of a hydrogen bond and 11 hydrophobic interactions (*Figure 4E*), whereas its binding to the active site of CHK1 is prevented by a steric clash with a tyrosine residue (Tyr86) (*Figure 4F*). To confirm the direct interaction between CHK2 and IBC, we performed the CETSA in MCF-7 cells. We showed that IBC altered the thermal stability of CHK2 as efficiently as

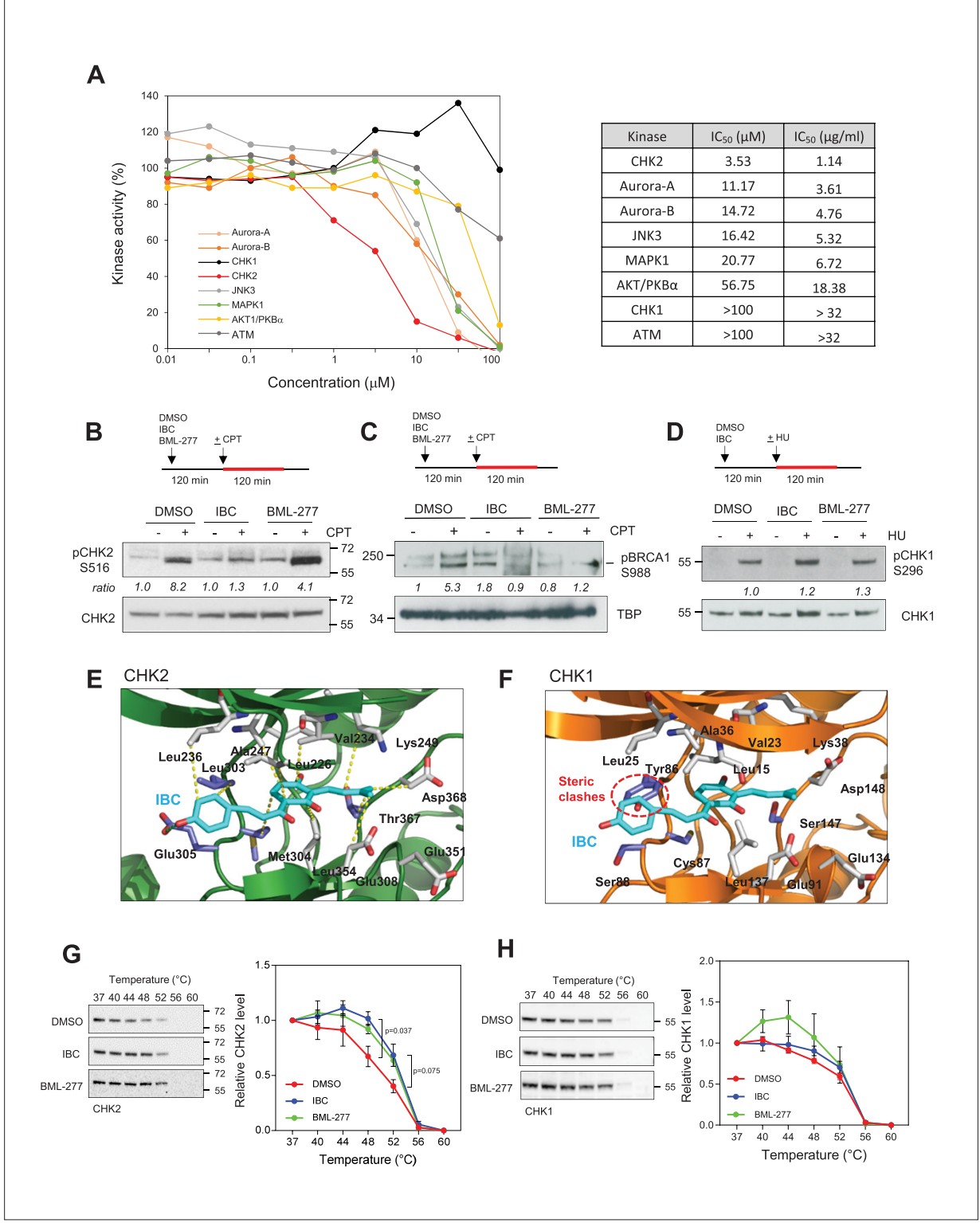

**Figure 4.** Isobavachalcone (IBC) inhibits the CHK2 kinase. (**A**) Selected protein kinases were incubated with the indicated range of IBC concentrations, and kinase activity in vitro was determined by using a radiometric assay. The IC$_{50}$ of IBC for each kinase is indicated in the panel on the right. (**B**) MCF-7 cells were pretreated with DMSO, 15 µM IBC, or 20 µM BML-277 for 2 hr, then camptothecin (CPT, 1 µM) was added for 2 hr. Phosphorylation of CHK2 on S516 (pCHK2) was detected by western blotting. The ratio of pCHK2-S516 induction, relative to the DMSO + CPT control, is indicated (n = 3). (**C**) MCF-7 cells were treated as indicated in (**B**). Phosphorylation of chromatin-bound BRCA1 at residue S988 was detected by western blotting. The relative ratio of pBRCA1-S988 signal, after normalization to Ponceau signal, is indicated. TBP was used as a marker of chromatin fraction. (**D**) MCF-7 cells

*Figure 4 continued on next page*

*Figure 4 continued*

were treated with 15 µM IBC for 2 hr, then 4 mM HU was added for 2 hr. CHK1 autophosphorylation on S296 (pCHK1) was detected by western blotting. (**E, F**) In silico molecular docking of IBC in the active sites of CHK2 and CHK1, respectively. (**G, H**) Cellular thermal shift assay (CETSA) of IBC on the thermal stability of CHK2 and CHK1. MCF-7 cells were treated with 15 µM IBC or 20 µMBML-277 for 2 hr. Cells were proceeded to CETSA as described in the 'Materials and methods'. The amount of CHK2 and CHK1 present in the supernatant was detected by western blotting. The relative CHK2 and CHK1 signal was quantified. The p-values were determined using two-tailed paired *t*-test (n = 3).

The online version of this article includes the following source data and figure supplement(s) for figure 4:

**Source data 1.** Original membranes corresponding to *Figure 4B–D, G, H* with labels.

**Source data 2.** Original membranes corresponding to *Figure 4B–D, G, H*.

**Figure supplement 1.** Isobavachalcone (IBC) does not inhibit AKT activity in MCF-7 cells.

**Figure supplement 1—source data 1.** Original membranes corresponding to *Figure 4—figure supplement 1*.

**Figure supplement 1—source data 2.** Original membranes corresponding to *Figure 4—figure supplement 1* with labels.

the commercial CHK2 inhibitor, BML-277, without affecting the thermal stability of CHK1 (*Figure 4G and H*). Interestingly, BML-277 seemed to have a slight effect in reducing the thermal stability of CHK1, although it was not statistically significant (*Figure 4G and H*). Together, these data indicate that IBC inhibits CHK2 without affecting CHK1 activity.

## IBC delays repair of DSBs induced by camptothecin

CHK2 promotes HR-mediated DSB repair (*Parameswaran et al., 2015*; *Zhang et al., 2004*). To investigate the effect of IBC on DSB repair, we induced chromosome breaks in MCF-7 cells by using CPT and monitored the persistence of unrepaired DSBs in the following $G_1$ phase by immunofluorescence microscopy of 53BP1 foci and co-staining with an antibody against p27, a marker of $G_1$ cells. In cells treated with IBC, the CPT-induced 53BP1 foci persisted, whereas in cells not treated with IBC, the intensity of the CPT-induced 53BP1 immunofluorescence signal increased and returned to basal levels 24 hr later (*Figure 5A and B*), indicating that IBC delays DSB repair. We also observed this persistence of DSBs in the presence of IBC by using pulsed-field gel electrophoresis (PFGE) (*Figure 5—figure supplement 1A*), confirming that IBC impairs DSB repair, likely by inhibiting CHK2.

## IBC prevents DNA end resection at DSBs

CHK2 phosphorylates BRCA1 on S988 to stimulate HR-mediated DSB repair (*Parameswaran et al., 2015*; *Zhang et al., 2004*). Since BRCA1 promotes DNA end resection at DSBs to initiate HR, we investigated whether IBC might impede the formation of single-strand DNA (ssDNA) at DNA ends. To do so, we induced DSBs by treating MCF-7 cells with CPT and assayed the formation of ssDNA by monitoring binding of the ssDNA-binding factor RPA to chromatin by western blotting and immunofluorescence microscopy. These analyses revealed that IBC inhibited the formation of RPA-coated ssDNA upon CPT treatment (*Figure 5C, Figure 5—figure supplement 1B*).

To determine whether this effect was due to inhibition of DNA end resection, we used single-molecule analysis of resection tracks (SMART) (*Cruz-García et al., 2014*). In this assay, the length of BrdU-labeled ssDNA tracks exposed by resection is measured after induction of DSBs in MCF-7 cells with bleomycin. We found that the BrdU tracks were significantly shorter when CHK2 was inhibited either by IBC or by the CHK2 inhibitor BML-277 (*Arienti et al., 2005*; *Figure 5D*). Using the same assay, we found that IBC also impaired DNA end resection induced by gamma irradiation (*Figure 5E*). Furthermore, we confirmed that IBC prevents DSB end resection by using the DIvA system, in which site-specific DSBs are generated by the restriction enzyme *AsiSI* to allow the quantification of ssDNA generated at break sites by quantitative PCR (*Iacovoni et al., 2010*). IBC inhibited the formation of ssDNA at two break sites induced by *AsiSI* digestion (*Figure 5—figure supplement 1C*, *Figure 5F*), indicating that inhibition of CHK2 by IBC inhibits DSB end resection. Moreover, the extent of inhibition by IBC was similar to that caused by BML-277 (*Figure 5—figure supplement 1C*).

To analyze the consequences of CHK2 inhibition by IBC on HR, we assayed formation of RAD51 foci following induction of DSBs by ionizing radiation (8 Gy) or by bleomycin treatment. CHK2 inhibition by either IBC or BML-277 completely prevented formation of RAD51 foci in response to ionizing radiation (*Figure 5G*) and IBC completely prevented formation of RAD51 foci in response to bleomycin

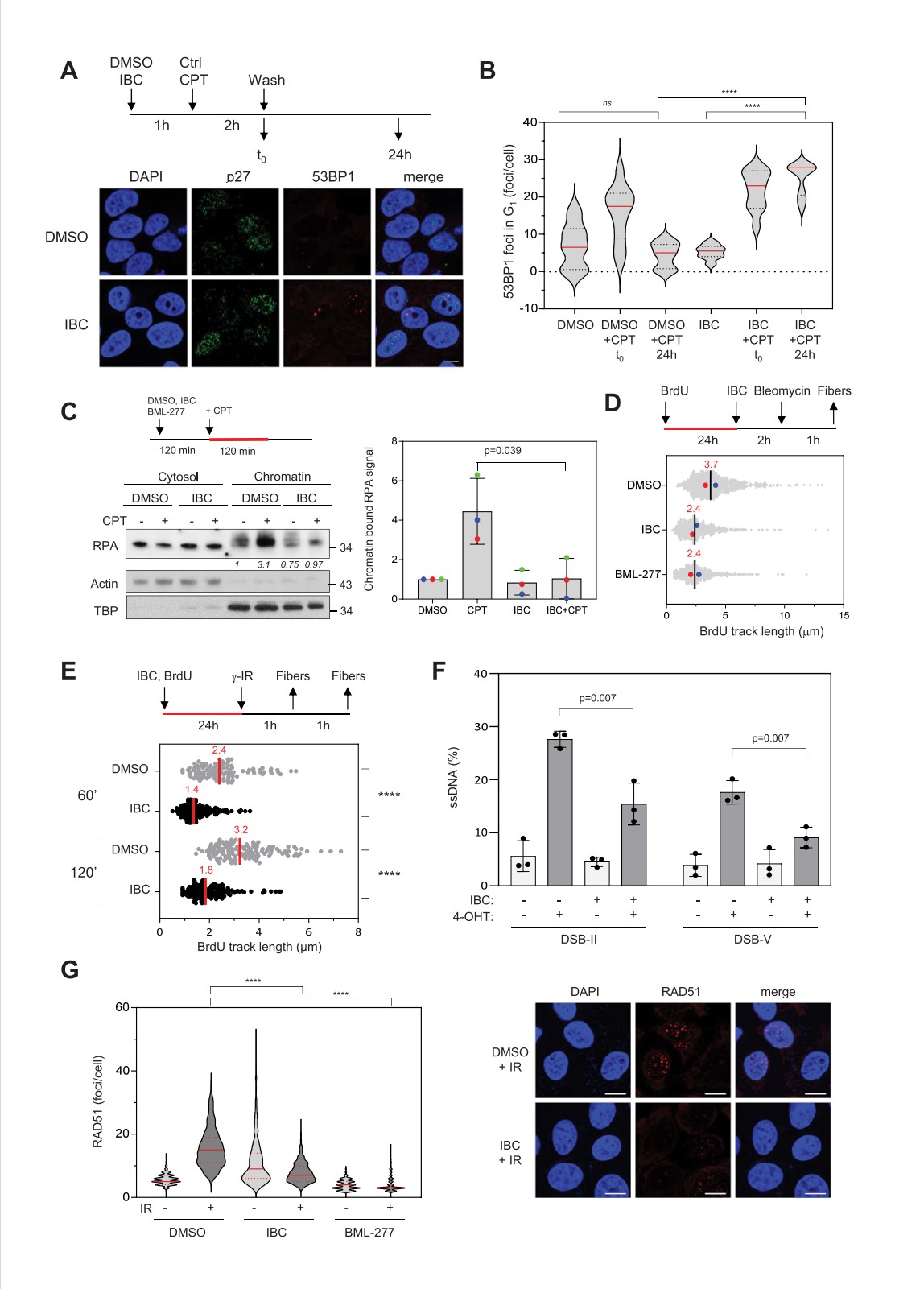

**Figure 5.** Isobavachalcone (IBC) impedes DNA end resection and RAD51 foci formation. (**A**) MCF-7 cells were treated with DMSO or 15 µM IBC for 1 hr, then camptothecin (CPT, 1 µM) was added for 2 hr. The cells were fixed immediately or washed and allowed to recover in medium containing DMSO or IBC for 24 hr before fixation. 53BP1 foci in p27-positive nuclei were detected by immunofluorescence microscopy. Representative images are shown. Bar: 5 µm. (**B**) 53BP1 foci number was quantified using CellProfiler. Representative data from three independent experiments are shown.

*Figure 5 continued on next page*

*Figure 5 continued*

****p<0.0001; *ns*, not significant, Mann–Whitney rank sum test. (**C**) MCF-7 cells were treated with DMSO or 15 µM IBC for 2 hr, then CPT (1 µM) was added for 2 hr, as indicated. Cells were fractionated into cytosol and nuclei (chromatin) and RPA in both fractions was detected by western blotting. Actin and TBP were used as markers of cytosol and chromatin fractions, respectively. The fold change of the chromatin-bound RPA signal relative to the DMSO control was quantified. The p-value was determined using unpaired *t*-test (n = 3). (**D**) MCF-7 cells were incubated with 10 µM BrdU for 24 hr to label genomic DNA, then DMSO, 15 µM IBC or the CHK2 inhibitor BML-277 (20 µM) were added for 2 hr followed by addition of 5 µg/ml bleomycin for 1 hr. DNA fibers were spread on glass slides and BrdU was detected by immunofluorescence microscopy without DNA denaturation. The length of BrdU the tracks was measured and the median for each condition is indicated in red. At least 250 fibers were measured for each condition. Median of two independent experiments is indicated in red (n = 2). (**E**) MCF-7 cells were incubated with 10 µM BrdU for 24 hr to label genomic DNA in the presence of DMSO or IBC then exposed to ionizing radiations (8 Gy). Cells were collected 60 or 120 min after irradiation and BrdU tracks measured as in (**B**). ****p<0.0001, Mann–Whitney rank sum test. (**F**) DIvA cells were treated with either DMSO or IBC for 2 hr then DNA breaks were induced by treatment with 300 nM 4-hydroxytamoxifen (4-OHT) for 4 hr. Resection at two break sites, DSB-II and DSB-V, was determined as the percentage of ssDNA at these sites, calculated as indicated in the 'Materials and methods'. Data are means ± SD (n = 3). The p-values are indicated (two-tailed paired *t*-test). (**G**) MCF-7 cells were treated with DMSO, IBC, or BML-277 for 2 hr, then irradiated as described above. After 1 hr, RAD51 foci were detected by CSK-immunofluorescence microscopy and foci number was quantified using CellProfiler. Representative data (left) and immunofluorescence images (right) from two independent experiments are shown. Bar: 10 µm. ****p<0.0001, Mann–Whitney rank sum test.

The online version of this article includes the following source data and figure supplement(s) for figure 5:

**Source data 1.** Original membranes corresponding to *Figure 5C* with labels.

**Source data 2.** Original membranes corresponding to *Figure 5C*.

**Figure supplement 1.** Isobavachalcone (IBC) results in the persistence of DNA breaks and impairs DNA end resection.

**Figure supplement 1—source data 1.** Original gel corresponding to *Figure 5—figure supplement 1A* with labels.

**Figure supplement 1—source data 2.** Original gel corresponding to *Figure 5—figure supplement 1A*.

---

(*Figure 5—figure supplement 1D*). Of note, the number of breaks induced by bleomycin was similar in cells treated with IBC or DMSO (*Figure 5—figure supplement 1E*).

## IBC and BKC synergistically inhibit tumor growth and extend survival in mice

To determine whether IBC and BKC might prevent cancer cell growth in vivo, we used a mouse xenograft model in which MCF-7 cells harboring an integrated firefly luciferase gene (MCF-7/Luc cells) were injected subcutaneously into the fat pads of non-obese diabetic/severe combined immunodeficiency (NOD/SCID) mice. Two days later, randomized mice were injected subsequently three times per week with two different concentrations of IBC, BKC, or IBC + BKC, or with Taxol as a positive control, or with PBS (phosphate-buffered saline) as a negative control. Tumor size was measured at intervals of 1 week up to 3 weeks by measuring bioluminescence intensity and survival rate was monitored daily up to 70 days (*Figure 6A*). After 3 weeks, IBC and BKC significantly inhibited tumor growth in a dose-dependent manner (*Figure 6B*, *Figure 6—figure supplement 1A*). IBC had a stronger effect on the growth of MCF-7/Luc cells than BKC had, but the combined use of both compounds had the greatest inhibitory effect on tumor growth (*Figure 6B*). Moreover, the mice injected with the higher dose of IBC or of the IBC + BKC combination survived longer than those treated with Taxol (*Figure 6C*).

To determine whether the tumor tissues had increased levels of RS or DNA damage when the mice were treated with IBC and BKC, we used immunohistochemistry to monitor cell proliferation (staining for Ki67) and checkpoint activation (staining for CHK1/CHK2 phosphorylation) and the TUNEL fluorescence assay for DNA fragmentation. Overall, IBC and BKC inhibited tumor cell proliferation (*Figure 6D*, *Figure 6—figure supplement 1B*) and induced DNA fragmentation, indicative of tumor cell death by apoptosis (*Figure 6E*, *Figure 6—figure supplement 1B*). Interestingly, IBC$_{5x}$ alone inhibited cell proliferation as efficiently as the IBC + BKC$_{5x}$ combination and better than Taxol (*Figure 6B*). However, the percentage of TUNEL-positive cells was much higher in mice treated with IBC + BKC$_{5x}$ than either drug alone (*Figure 6E*), indicating that IBC and BKC have a strong synergistic effect on the induction of apoptosis and DNA breaks. We also observed a dose-dependent increase in CHK1 phosphorylation in tumor tissues from mice treated with BKC (*Figure 6F*), which is consistent with our observation above that BKC induced CHK1 activation in vitro. Moreover, BKC induced phosphorylation of CHK2 on T68, a marker of DNA damage (*Figure 6G*). However, this activation of CHK1 and CHK2 mediated by BKC was suppressed by the addition of IBC (*Figure 6F and G*). Since these checkpoint kinases are important to coordinate DNA repair, this would explain why unrepairable DNA

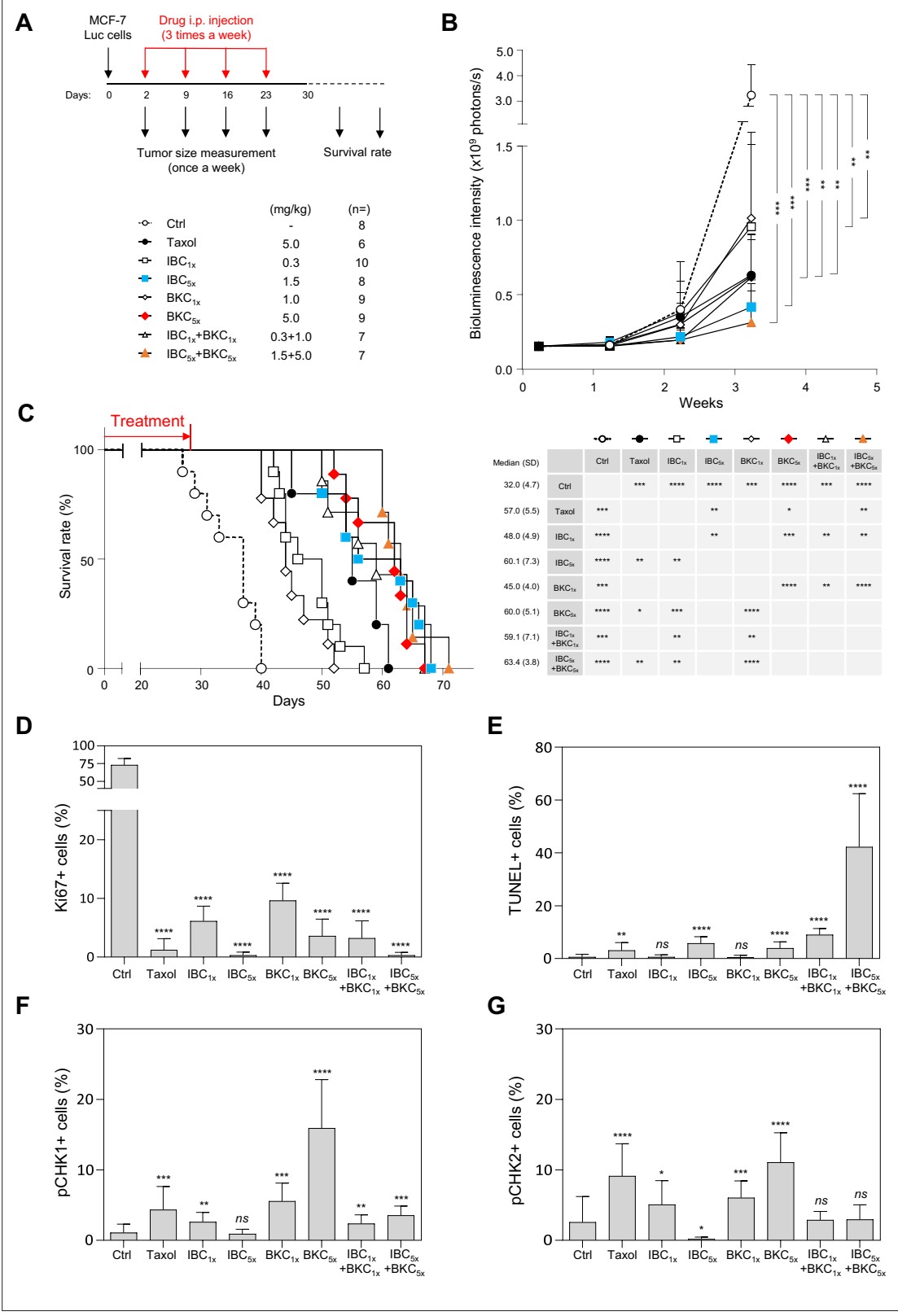

**Figure 6.** Isobavachalcone (IBC) and bakuchiol (BKC) synergistically inhibit tumor development and induce DNA damage in a xenograft mouse model. (**A**) Schematic description of the protocol. MCF-7/Luc cells (1 × 10⁴) were injected into the fat pads of female non-obese diabetic/severe combined immunodeficiency mice at day 0. Two days later, randomized mice were injected intraperitoneally with (phosphate-buffered saline) PBS, Taxol, IBC, BKC, or IBC + BKC at the indicated doses. Tumor sizes were measured weekly thereafter. Tumor tissues were collected 28 days after grafting and analyzed by

*Figure 6 continued on next page*

*Figure 6 continued*

immunohistochemistry. Mice survival was also evaluated. (**B**) Tumor size in the fat pads was measured once per week by using the IVIS bioluminescence system. The number of xenografted mice receiving each treatment is indicated. ****p<0.0001, ***p<0.001, **p<0.01, *p<0.05, Mann–Whitney rank sum test. (**C**) Survival (left) and median survival time (days, right) is shown for xenografted mice receiving each treatment. ****p<0.0001, ***p<0.001, **p<0.01, *p<0.05, Mann–Whitney rank sum test. (**D**) Tumor tissues collected from xenograft mice were analyzed immunohistochemically for the cell proliferation maker Ki67. (**E**) Broken DNA and apoptosis marker, TUNEL staining. (**F**) pCHK1 (S345) and (**G**) pCHK2 (Thr68). ****p<0.0001, ***p<0.001, **p<0.01, *p<0.05; *ns*, not significant, Mann–Whitney rank sum test.

The online version of this article includes the following figure supplement(s) for figure 6:

**Figure supplement 1.** Isobavachalcone (IBC) and bakuchiol (BKC) extend survival and induce replication stress and DNA damage in a xenograft mouse model.

breaks accumulate in tumors exposed to the IBC + BKC$_{5x}$ combination. Interestingly, treatment with the higher concentration of IBC alone diminished the level of endogenous phosphorylation of both CHK1 and CHK2, which supports the view that IBC prevents the signaling of endogenous RS and DNA damage in tumors (*Figure 6F and G*). Altogether, our results show that IBC and BKC act synergistically to inhibit tumor growth in vivo, induce DNA fragmentation in the tumor, and extend mice survival.

## IBC potentiates the effect of chemotherapeutic agents on lymphoma cells

One current strategy for new cancer treatments is to inhibit cell cycle checkpoints at the same time as inducing DNA damage with conventional chemotherapeutic agents, thus driving cells to proliferate in the presence of DNA damage, ultimately resulting in their death. We showed above that IBC inhibits the DNA damage checkpoint kinase CHK2; therefore, we investigated whether IBC might enhance the potency of chemotherapeutic drugs. To this end, we used cell lines from patients with DLBCL; this is the most common lymphoid malignancy in adults, accounting for up to 35% of non-Hodgkin lymphomas. Although DLBCL can be cured in over 60% of patients by using rituximab-based chemotherapy regimens, the remainder develop recurrent or progressive disease that is often fatal (*Sarkozy and Coiffier, 2013*). New therapeutic approaches are still needed to achieve an effective treatment for these patients with high-risk/refractory DLBCL. Since deregulation of DNA repair pathways in DLBCL cells is associated with a poor outcome (*Bret et al., 2015*; *Bret et al., 2013*), we reasoned that IBC could potentiate the effect of agents inducing DNA damage in DLBCL cells.

To address this possibility, we first determined the IC$_{50}$ of IBC for growth inhibition of a panel of DLBCL cell lines. IC$_{50}$ ranged from 8 to 28 µM (*Figure 7A*), similar to the concentrations we found were effective on the solid cancer cell lines we tested (*Figure 1B and C*). To evaluate whether IBC potentiates the growth inhibitory effect of chemotherapeutic agents, we treated the drug-resistant DLBCL cell line U2932 with various concentrations of including etoposide and doxorubicin (topoisomerase II inhibitors) or 4-hydroxy-cyclophosphamide (DNA alkylating agent) in the presence of the IC$_{20}$ of IBC (4.5 µM) for 72 hr. We found that IBC substantially enhanced cell growth inhibition by all three of these DNA-damaging agents (*Figure 7B*). Moreover, by testing a full-range concentration matrix of drug pairs on cell viability after 72 hr treatment, we found that IBC synergistically increased the inhibitory effect of doxorubicin and or 4-hydroxy-cyclophosphamide on U2932 cell proliferation (*Figure 7C*, *Figure 7—figure supplement 1*). These matrices were more complex than those shown in *Figure 1C* for BKC, presumably because Top2 inhibition or DNA alkylation induce pleiotropic effects differentially modulating the effect of IBC. Nonetheless, these data suggest that IBC is a promising candidate to potentiate the effect of these and potentially other conventional chemotherapeutic agents.

## Discussion

Approaches using small-molecule inhibitors of cell cycle checkpoint kinases have shown promise as antitumor agents in preclinical studies, either when used alone or in combination with genotoxic agents that induce RS. However, clinical trials with these agents have been disappointing, showing only limited benefit to patients and significant side effects, particularly in combination therapies (*Zhu et al., 2020*). Here, we sought to identify novel small-molecule inhibitors that sensitize cancer cells to RS but are not toxic to noncancer cells. By screening a collection of plant extracts used in traditional Chinese medicine, we identified two known compounds, BKC and IBC, from *P. corylifolia*, that

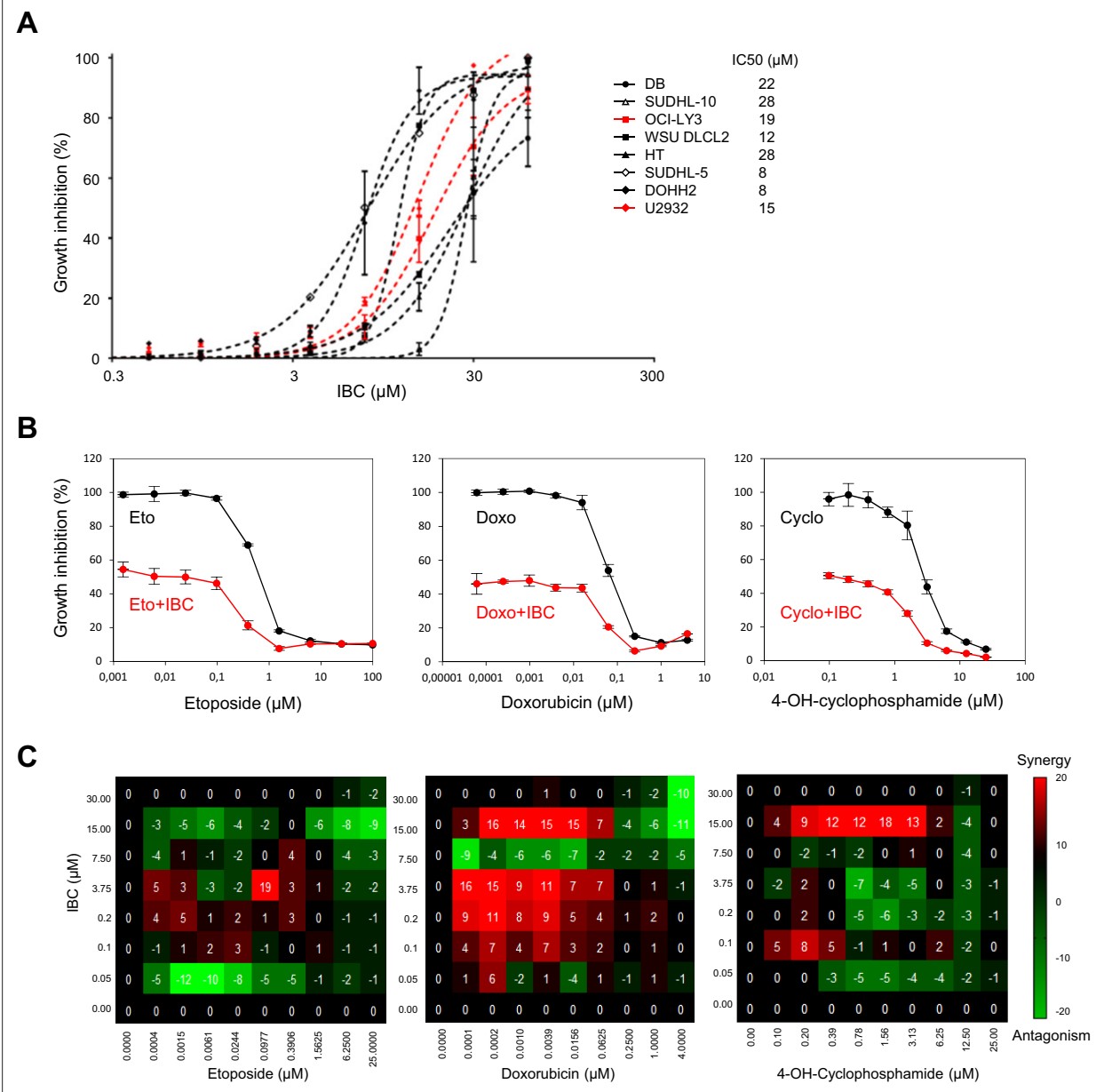

**Figure 7.** Isobavachalcone (IBC) potentiates the effect of chemotherapeutic agents on lymphoma cells. (**A**) A panel of eight diffuse large B-cell lymphoma (DLBCL) cell lines were incubated with the indicated concentrations of IBC for 72 hr (left) and growth inhibition was measured to calculate the $IC_{50}$ in each cell line (right). Data are presented as means ± SD (n = 3). (**B**) U2932 cells were treated with the indicated concentrations of etoposide (Eto), doxorubicin (Doxo), or 4-OH-cyclophosphamide (4-OH-Cyclo) without (black points) or with 1.5 µg/ml (the $IC_{20}$ concentration for this cell line) IBC (red points) for 72 hr. Cell viability was measured. Data are presented as means ± SD (n = 3). (**C**) Full-concentration matrix analyses of U2932 cells treated with IBC and Eto (left), Doxo (middle), or 4-OH-Cyclo at the indicated concentrations for 72 hr. Synergy and antagonism were calculated as described in *Figure 1*.

The online version of this article includes the following figure supplement(s) for figure 7:

**Figure supplement 1.** Isobavachalcone (IBC) potentiates the anticancer of chemotherapeutic agents in diffuse large B-cell lymphoma (DLBCL).

synergistically inhibit proliferation of cancer cells. We show that BKC inhibits DNA replication, likely by binding to DNA polymerases, and that IBC prevents DNA end resection by inhibiting the checkpoint kinase CHK2. Together, these drugs induce RS and impede HR-mediated DSB repair. Consistent with these effects, we show that the drugs act synergistically to inhibit tumor development and extend survival rate in a xenograft mouse model. Moreover, IBC potentiates the inhibitory effect of conventional chemotherapeutic agents on lymphoma cell lines.

BKC is similar in structure to resveratrol and has been reported to exert a variety of pharmacological effects (*Xin et al., 2019*). In particular, BKC inhibits the proliferation of lung, breast, skin, and stomach cancer cell lines (*Kim et al., 2016*; *Li et al., 2016*; *Lv and Liu, 2017*) and prevents replication of the polyomavirus SV40 in vitro (*Sun et al., 1998*). This is consistent with our evidence from in silico molecular docking that BKC binds the catalytic site of DNA polymerases δ and ε. Thus, BKC probably inhibits the polymerases by competing for dNTP binding. Indeed, we demonstrate that BKC inhibits replication elongation of chromosomal DNA and ssDNA in *Xenopus* egg extracts, albeit to a lesser extent than does aphidicolin. These findings argue against the possible explanation that BKC inhibits dNTP synthesis, as proposed for the structurally similar compound resveratrol (*Benslimane et al., 2020*; *Fontecave et al., 1998*), since dNTPs are available in large amount in these extracts. Moreover, we show that BKC slows replication fork progression in vivo in both normal and cancer cells, indicating that it acts as a bona fide replication inhibitor.

IBC affects a wide spectrum of biological functions (*Kuete and Sandjo, 2012*), including the activity of the NAD+-dependent deacetylase Sirtuin 2 (*Ren et al., 2024*) and shows antitumor activity against drug-resistant cancers (*Kuete et al., 2015*; *Ren et al., 2024*; *Wu et al., 2022*). IBC is thought to prevent cell proliferation and induce apoptosis in various cancer cell models by inhibiting the AKT kinase (*Jin and Shi, 2016*; *Jing et al., 2010*). In our hands, however, IBC does not significantly inhibit AKT, at least when the drug was used at concentrations that inhibit cell proliferation. Rather, IBC inhibits CHK2 in vivo and in vitro at concentrations much lower than those that inhibit other kinases and it has no effect on the activity of the related checkpoint kinase CHK1. As a key effector of the DNA damage response, CHK2 is an attractive target for new drugs that might potentiate the effect of conventional, DNA-damaging treatments for cancer (*Bucher and Britten, 2008*). However, few CHK2-specific inhibitors are available and those that are have only modest antiproliferative effects compared with inhibitors of other checkpoint kinases such as CHK1 (*Ronco et al., 2017*). The results of clinical trials combining the CHK2 inhibitor PHI-101 with other therapeutic agents are still awaited (*Park et al., 2022*). Interestingly, CHK2 inhibitors also protects healthy tissues from radiotherapy or chemotherapy, presumably by inhibiting p53-dependent apoptosis (*Jiang et al., 2009*; *Xu et al., 2021*).

The target of IBC in vivo is very likely CHK2-dependent activation of BRCA1. Indeed, we show that IBC prevents HR-mediated DSB repair by inhibiting the formation of RAD51 foci in cancer cells. The loading of RAD51 depends on the resection of DNA ends by nucleases, through a process that is mediated by the CHK2-dependent activation of BRCA1 (*Parameswaran et al., 2015*; *Zhang et al., 2004*). We found that IBC inhibits resection at DSBs as efficiently as the CHK2 inhibitor BML-277. In contrast, IBC does not inhibit resection of nascent DNA at stalled forks, a process that does not depend on BRCA1 but, rather, is repressed by BRCA1 and BRCA2 (*Chen et al., 2018*). Our data indicate therefore that IBC differentially affects resection at DSB and at stalled forks by inhibiting CHK2.

The inhibitory effects of BKC on DNA synthesis and IBC on DNA end resection support a model in which BKC increases RS in cancer cells by inhibiting bulk DNA synthesis, so increasing the rate of

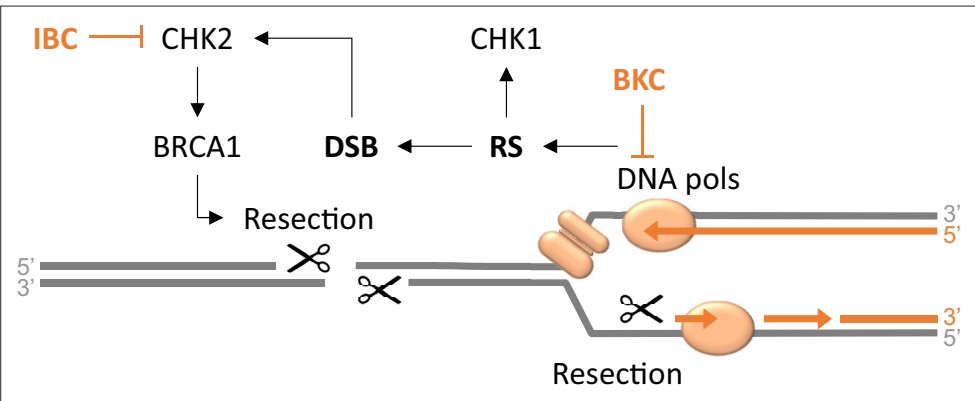

**Figure 8.** Model of the synergistic action of bakuchiol (BKC) and isobavachalcone (IBC) that kills cancer cells. By inhibiting the DNA polymerases, BKC enhances the endogenous replication stress (RS) in cancer cells, leading to increased DNA damage. IBC specifically inhibits CHK2 to prevent DNA repair. The combined use of BKC and IBC enhances DNA damage to a level that triggers cancer cell death.

replication fork arrest/collapse and inducing RS-dependent DSBs, and IBC inhibits HR-mediated DSB repair, resulting in cell death by apoptosis (*Figure 8*). This model explains why BKC and IBC synergistically inhibit the proliferation of cancer cells: BKC-mediated RS adds to oncogene-induced RS, explaining the differential sensitivity of normal and cancer cells to the combination of BKC and IBC, and also to the effects of IBC alone.

In conclusion, our screening of traditional herbal medicine for novel combinations of small-molecule inhibitors identified two compounds, BKC and IBC, acting synergistically to block cancer cell growth in different in vitro and in vivo contexts. Although it is unlikely that these compounds will ever be used in combination therapies, our results indicate that IBC or other clinically approved CHK2 inhibitors could be used to potentiate the cytotoxic effect of known DNA damaging agents. This view is supported by our data showing that IBC potentiates the effect of etoposide, doxorubicin, or 4-hydroxy-cyclophosphamide on drug-resistant lymphoma cell lines. Finally, recent evidence indicates that RS and DNA damage stimulate anti-tumor immunity by activating cytosolic nucleic acid-sensing pathways (*Chabanon et al., 2021*; *Tarsounas and Sung, 2020*; *Técher and Pasero, 2021*). New combinations of genotoxic agents and inhibitors targeting RS checkpoint kinases might thus be valuable also for potentiating immune checkpoint inhibitors.

## Materials and methods

**Key resources table**

| Reagent type (species) or resource | Designation | Source or reference | Identifiers | Additional information |
|---|---|---|---|---|
| Cell line (*Homo sapiens*) | Immortalized BJ fibroblasts | Dr. D. Peeper | The Netherlands Cancer Institute, Amsterdam | Foreskin (normal neonatal mal) |
| Cell line (*H. sapiens*) | MCF-7 | HTB-22 | ATCC | Mammary gland adenocarcinoma |
| Cell line (*H. sapiens*) | A549 | CRM-CCL-185 | ATCC | Lung carcinoma |
| Cell line (*H. sapiens*) | HCC827 | CRL-2868 | ATCC | Lung adenocarcinoma |
| Cell line (*H. sapiens*) | PC3 | CRL-1435 | ATCC | Prostate adenocarcinoma |
| Cell line (*H. sapiens*) | DU145 | HTB-81 | ATCC | Prostate carcinoma |
| Cell line (*H. sapiens*) | U937 | CRL-1593.2 | ATCC | Histiocytic lymphoma |
| Cell line (*H. sapiens*) | HCT116 | CCL-247 | ATCC | Colorectal carcinoma |
| Cell line (*H. sapiens*) | OVCAR8 | NIH:OVCAR8 | | Ovarian carcinoma |
| Cell line (*H. sapiens*) | DLBCL cell lines | Dr. J. Moreaux | Institute of Human Genetics, Montpellier | |
| Cell line (*H. sapiens*) | SUM159 | SUM159PT | Asterand Bioscience | Triple-negative breast cancer cell line |
| Antibody | Mouse monoclonal anti-BrdU clone B44 | 347580 | BD Biosciences | 1/100 |
| Antibody | Rat monoclonal anti-BrdU clone BU1/75 | ABC117-7513 | Eurobio Abcys | 1/100 |
| Antibody | Mouse monoclonal anti-ssDNA | MAB3868 | Millipore | 1/250 |
| Antibody | Rabbit monoclonal anti-pCHK1 (S345) | 2348 | Cell Signaling | 1/1000 |
| Antibody | Rabbit polyclonal anti-pCHK2 (T68) | 2661 | Cell Signaling | 1/1000 |
| Antibody | Mouse monoclonal anti-γ-H2AX (S139) | 05-636 | Millipore | 1/500 |
| Antibody | Mouse monoclonal anti-actin | A4700 | Sigma | 1/500 |
| Antibody | Rabbit monoclonal anti-RPA1 | Ab79398 | Abcam | 1/300 |

*Continued on next page*

*Continued*

| Reagent type (species) or resource | Designation | Source or reference | Identifiers | Additional information |
|---|---|---|---|---|
| Antibody | Rabbit polyclonal anti-TBP | 8515 | Cell Signaling | 1/1000 |
| Antibody | Mouse monoclonal anti-CHK1 | 2360 | Cell Signaling | 1/1000 |
| Antibody | Rabbit polyclonal anti-pRPA32 (S4/S8) | A300-245A | Bethyl | 1/1000 |
| Antibody | Rabbit monoclonal anti-CHK2 | ab109413 | Abcam | 1/5000 |
| Antibody | Goat polyclonal Anti-RPA2 | A303-874 | Bethyl | 1/500 |
| Antibody | Rabbit polyclonal anti-pCHK2 (S516) | 2669 | Cell Signaling | 1/1000 |
| Antibody | Rabbit polyclonal anti-pChk1 (S296) | 2349 | Cell Signaling | 1/1000 |
| Antibody | Mouse monoclonal anti-pBRCA1 (S988) | sc-166793 | Santa Cruz | 1/200 |
| Antibody | Rabbit polyclonal anti-RAD51 | PC130 | Millipore | 1/500 |
| Antibody | Rabbit polyclonal anti-pAKT (S473) | 9271 | Cell Signaling | 1/1000 |

## Cell culture

Immortalized human BJ fibroblasts and MCF-7/Luc were cultured in Dulbecco's modified Eagle's medium (DMEM) supplemented with 10% fetal calf serum and 100 U/ml penicillin/streptomycin. SUM159 cells were obtained from Asterand Bioscience, UK, and grown in Ham's F-12 medium supplemented with 5% fetal calf serum, 10 µg/ml insulin, 1 µg/ml hydrocortisone, 100 µg/ml streptomycin, and 100 U/ml penicillin. MCF-7, A549, HCC827, PC3, DU145, U937, HCT116, and OVCAR8 cancer cells were cultured in DMEM (MCF-7, A549, DU145, and OVCAR8) or RPMI-1640 (HCC827, PC3, U937, HCT116) supplemented with 10% fetal calf serum and 100 U/ml penicillin/streptomycin at 37°C in 5% $CO_2$. DLBCL cell lines (DB, SUDHL-10, OCI-LY3, WSU DLCL2, HT, SUDHL-5, SOHH2, and U292) were cultures in RMPI-1640 supplemented with 10% fetal calf serum and 100 U/ml penicillin/streptomycin at 37°C in 5% $CO_2$.

## Reagents

Isobavachalcone was purchased from Sigma-Aldrich (SML1450) or Abcam (ab141168). Bakuchiol was from Abcam (ab141036). Cell proliferation reagent WST-1 was from Sigma-Aldrich (5015944001).

## Pulse-field gel electrophoresis (PFGE)

Subconfluent cultures (10 cm plates) were treated as specified. Cells were harvested by trypsinization, and plugs of 2% (w/v) agarose containing $0.5 \times 10^6$ cells in PBS were prepared using a CHEF disposable plug mold (Bio-Rad). The plugs were incubated in lysis buffer (100 mM EDTA, 1% [w/v] sodium lauryl sarcosinate, 0.2% [w/v] sodium deoxycholate, 1 mg/ml proteinase K) at 37°C for 24 hr and were then washed with washing buffer (20 mM Tris pH8, 50 mM EDTA pH8). PFGE was carried out at 13°C for 23 hr in 0.9% (w/v) agarose containing 0.25% TBE buffer using a Biometra Rotaphor (Biometra). The parameters were as follows: voltage 180–120 V log; angle from 120° to 110° linear; interval 30–5 s log. The gel was stained with ethidium bromide (EtBr) and analyzed using ImageJ.

## Flow cytometry analysis

Cells were pulse labeled with 10 µM EdU for 30 min. After fixation with 1% formaldehyde for 30 min at room temperature and permeabilized in 0.25% Triton X-100 for 15 min, EdU incorporation was detected by using Click chemistry according to the manufacturer's instructions (Click-iT EdU Flow Cytometry Cell Proliferation Assay, Invitrogen). The cells were resuspended in PBS containing 1% (w/v) BSA, 2 µg/ml DAPI, and 0.5 mg/ml RNase A for 30 min at room temperature and were analyzed in a

MACSQuant flow cytometer (Miltenyi Biotec). The percentages of cells in $G_1$, S and $G_2$/M phases were quantified using FlowJo single-cell analysis software (FlowJo, LLC).

## DNA fiber spreading

DNA fiber spreading was performed as described previously (*Coquel et al., 2018*; *Jackson and Pombo, 1998*). Briefly, subconfluent cells were sequentially labeled first with 10 μM IdU and then with 100 μM CldU for the indicated times. 1000 cells were loaded onto a glass slide (StarFrost) and lysed with spreading buffer (200 mM Tris-HCl pH 7.5, 50 mM EDTA, 0.5% SDS) by gently stirring with a pipette tip. The slides were tilted slightly and the surface tension of the drops was disrupted with a pipette tip. The drops were allowed to run down the slides slowly, then air dried, fixed in methanol/acetic acid 3:1 for 10 min, and allowed to dry. Glass slides were processed for immunostaining with mouse anti-BrdU to detect IdU, rat anti-BrdU to detect CldU, mouse anti-ssDNA antibodies (see Supplemental Information for details) and corresponding secondary antibodies conjugated to various Alexa Fluor dyes. Nascent DNA fibers were visualized by using immunofluorescence microscopy (Leica DM6000 or Zeiss ApoTome). The acquired DNA fiber images were analyzed by using MetaMorph Microscopy Automation and Image Analysis Software (Molecular Devices), and statistical analysis was performed with GraphPad Prism (GraphPad Software). The lengths of at least 150 IdU and/or CldU tracks were measured per sample.

## Single-molecule analysis of resection tracks (SMART)

Cells were labeled with 10 μM BrdU for 24 hr. They were then treated with 5 μM bleomycin (Calbiochem) for 1 hr and harvested at the indicated time points. They were processed for DNA fiber spreading as described (*Altieri et al., 2020*; *Cruz-García et al., 2014*). BrdU tracks were stained with anti-BrdU antibody without DNA denaturation and visualized by fluorescence microscopy (Zeiss ApoTome). The acquired DNA fiber images were analyzed by using MetaMorph Microscopy Automation and Image Analysis Software (Molecular Devices), and statistical analysis was performed with GraphPad Prism (GraphPad Software). The lengths of at least 200 BrdU tracks were measured per sample.

## Cellular thermal shift assay (CETSA)

CETSA was performed according to the protocol described by *Delport and Hewer, 2022* with the following modifications. MCF-7 cells were treated with DMSO, IBC, or BML-277 for 2 hr, then trypsinized and pelleted. Cells were resuspended in PBS and evenly distributed into different Eppendorf tubes at $10^6$ cells per tube. Tubes were incubated at specified temperatures as indicated for 3 min, cooled down at 25°C for 3 min and then incubated on ice for another 3 min. Cells were pelleted down by centrifugation. The pellets were then lysed with RIPA buffer in the presence of benzonase for 20 min at 4°C before further centrifugation. The supernatant was collected and subjected to SDS-PAGE before western blotting for the detection of CHK1 and CHK2.

## RPA foci detection

For the detection of chromatin-bound RPA foci, cells seeded on the coverslips were fixed with 4% PFA (Paraformaldehyde) in PBS for 15 min and then incubated for 3 min at 4°C with CSK buffer (10 mM PIPES pH 6.8, 100 mM NaCl, 1 mM $MgCl_2$, 1 mM EGTA, 300 mM sucrose, 0.5 mM DTT) containing 0.25% Triton X-100 and phosphatase inhibitor cocktail (Sigma-Aldrich, P0044). The coverslips were incubated with an anti-RPA antibody (overnight at 4°C) and then with a secondary antibody conjugated to an Alexa Fluor dye for 1 hr at 37°C, followed by DAPI staining. Images were acquired by using a Zeiss ApoTome microscope. The mean fluorescence intensity in EdU-positive cells was quantified by using CellProfiler (http://www.cellprofiler.org).

## γ-H2AX foci detection

Cells seeded on coverslips were treated and labeled with 10 μM EdU for 10 min as described. They were washed twice with PBS. They were fixed in fixation buffer (2% PFA) for 10 min at room temperature and permeabilized in permeabilization buffer (0.1% Na citrate, 0.1% Triton X-100). The coverslips were incubated with an anti-γ-H2AX antibody overnight at 4°C after blocking in PBS containing 1% BSA in PBS for 1 hr at room temperature. Coverslips were incubated with a secondary antibody conjugated to Alexa Fluor dye, followed by Click chemistry reaction and DAPI staining. Images were

acquired using a Zeiss ApoTome microscope. The mean fluorescence intensity (MFI) in EdU-positive cells was quantified using CellProfiler (http://www.cellprofiler.org).

## DNA end resection assay

Measure of resection was performed as described previously with the following modifications (*Zhou et al., 2014*). Genomic DNA was extracted from fresh cells using the QIAamp mini kit (QIAGEN). 500 ng DNA was then treated with five units of RNase H. 200 ng RNase H-treated DNA were digested or not with the Ban I restriction enzyme (16 U per sample) overnight at 37°C, which cuts at ~200 bp from the DSB-KDELR3 and at 740 bp for DSB-ASXL1. Ban1 was heat inactivated 20 min at 65°C. Digested and undigested DNA were analyzed by qPCR using the following primers:

> DSB-KDELR3_200 FW: ACCATGAACGTGTTCCGAAT;
> DSB-KDELR3_200_REV: GAGCTCCGCAAAGTTTCAAG;
> DSB-ASXL1_740 FW: GTCCCCTCCCCCACTATTT;
> DSB-ASXL1_740_REV: ACGCACCTGGTTTAGATTGG; ssDNA% was calculated with the following equation: ssDNA% = $1/(2^{(Ct\ digested-Ct\ undigested-1)} + 0.5)*100$.

## In vitro kinase assay

Kinase selectivity was evaluated using KinaseProfiler service provided by Eurofins (https://www.eurofinsdiscovery.com/).

## DNA replication assay using *Xenopus* egg extracts

Cytoplasmic extracts (low speed and high speed) and demembranated sperm nuclei were prepared as previously described (*Méchali and Harland, 1982*; *Murray, 1991*), snap frozen in liquid nitrogen and stored at –80°C. Upon thawing, extracts were supplemented with cycloheximide (250 µg/ ml) and an energy regeneration system (1 mM ATP, 2 mM MgCl$_2$, 10 µg/ml creatine kinase, 10 mM creatine phosphate).

Egg extracts were supplemented with α-[32P] dATP (3000 Ci/mmol, PerkinElmer) and either demembranated sperm nuclei (1000/l of extract) or M13 ssDNA (200 ng/µl; NEB). At the indicated time points, samples were neutralized in 10 mM EDTA, 0.5% SDS, 200 µg/ml Proteinase K (Sigma) and incubated at 37°C overnight. Incorporation of radioactive label was determined by TCA precipitation on GF/C glass fiber filters (Whatman) following by scintillation counting.

## Full-range dose matrix approach

To investigate the interactions between two-drug combinations, we used a synergy matrix assay that was previously described in details (*Tosi et al., 2018*). The effects of drug combinations on cell growth was evaluated by standard sulforhodamine B (SRB) assay as described (*Orellana and Kasinski, 2016*). Briefly, exponentially growing cells were treated with all the combinations of five concentrations of BKC and eight concentrations of IBC in 96-well plates for 72 hr. Cells were then fixed with trichloroacetic acid solution (10%) and stained with a 0.4% sulforhodamine B solution in 1% acetic acid, washed with 1% acetic acid and incubated with 10 mM Tris-HCl solution for 10 min with gentle shaking. Absorbance at 560 nm was then measured using a PHERAstar FS plate reader (BMG Labtech, Ortenberg, Germany), and cell survival (blue matrix) was calculated in comparison with untreated cells. Experiments have been performed three times independently for each cell line, and a representative matrix is shown as an example for each cell line. A synergy matrix was then calculated as described previously (*Tosi et al., 2018*) to quantify the interaction effect: a red color in the matrix indicates a synergism, a black color additivity, and a green color an antagonism.

## In silico molecular docking

To explore human DNA polymerase and BKC interaction, BKC was docked into the polymerase active site using GOLD docking tool with BIOVIA Discovery Studio (Dassault Systèmes, BIOVIA Corp., San Diego, CA). In order to construct human polymerase delta and epsilon protein models, we carried out homology modeling using yeast polymerase delta (PDB code 3IAY) and epsilon from *Saccharomyces cerevisiae* (PDB code 4M8O) as templates, respectively. Similarly, to investigate the interaction between checkpoint kinase and IBC, IBC was docked into the active site of CHK1 (PDB code 5F4N)

and CHK2 (PDB code 4BDK). The crystal structures of DNA polymerase and checkpoint kinase were downloaded from RCSB Protein Data Bank. The proteins and compound atoms were applied with CHARMm force field.

## Animal study

Female NOD/SCID mice were National Laboratory Animal Center, Taiwan. All mice had free access to chow and water, and were housed at 21–23°C with 12 hr light–12 hr dark cycles. All mice were handled in accordance with the guidelines laid out by the Academia Sinica Institutional Animal Care and Utilization Committee (protocol no. 10-12-097). To generate MCF tumor-bearing mice, MCF-7/Luc cells ($1 \times 10^4$) were subcutaneously inoculated into the fat pad of the mice as published (*Kuo et al., 2017*). Eight groups of the mice received an intraperitoneal injection of PBS (Ctl), Taxol (5 mg/kg), IBC (0.3 and 1.5 mg/kg), BKC (1 and 5 mg/kg), and a combination of IBC and BKC (0.3 mg/kg IBC + 1 mg/kg BKC and 1.5 mg/kg IBC + 5 mg/kg BKC), thrice a week, from days 2 to 30 after tumor graft. The mice were daily measured for survival rate. Their tumor growth was weekly monitored using the IVIS system (Xenogen, USA). The signal of the bioluminescence from mice was quantified using Living Image 2.5 (Xenogen) as photons/s/region of interest.

## Immunohistochemical analysis

Tumors were removed from mice 28 days post tumor inoculation. The tumors were fixed, dehydrated, and embedded into paraffin. The tumor sections were stained with the antibody against Ki67, p-CHK1, and p-CHK2 and TUNEL kits. The signal of the sections was visualized and quantified using AxioVision software (Carl Zeiss MicroImaging).

# Acknowledgements

We thank H Técher, D Gopaul, and B Pardo for comments on the manuscript and C Featherstone (Plume Scientific Communication Services) for professional editing. This work was supported by grants from the Institut National du Cancer (INCa) and La Ligue Contre le Cancer (équipe labelisée) to PP, and from the Fondation ARC pour la Recherche Contre le Cancer to YLL, and from the Programme de Pré-maturation, Région Occitanie to DM.

---

# Additional information

## Funding

| Funder | Grant reference number | Author |
|---|---|---|
| Institut National Du Cancer | | Philippe Pasero |
| Fondation ARC pour la Recherche sur le Cancer | | Yea-Lih Lin |
| Ligue Contre le Cancer | | Philippe Pasero |
| Programme de Pré-maturation, Région Occitanie | | Domenico Maiorano |

The funders had no role in study design, data collection and interpretation, or the decision to submit the work for publication.

## Author contributions

Flavie Coquel, Chun-Yen Yang, Investigation, Methodology; Sing-Zong Ho, Methodology; Keng-Chang Tsai, Software, Methodology; Antoine Aze, Julie Devin, Ting-Hsiang Chang, Marie Kong-Hap, Audrey Bioteau, Investigation; Jerome Moreaux, Domenico Maiorano, Philippe Pourquier, Supervision, Methodology; Wen-Chin Yang, Conceptualization, Resources, Supervision, Methodology; Yea-Lih Lin, Conceptualization, Resources, Supervision, Funding acquisition, Methodology, Writing – original draft, Writing – review and editing; Philippe Pasero, Conceptualization, Supervision, Funding acquisition, Writing – original draft, Writing – review and editing

## Author ORCIDs
Sing-Zong Ho (iD) https://orcid.org/0009-0008-9986-6061
Keng-Chang Tsai (iD) https://orcid.org/0000-0001-8277-9174
Jerome Moreaux (iD) https://orcid.org/0000-0002-5717-3207
Domenico Maiorano (iD) https://orcid.org/0000-0003-4229-5903
Philippe Pourquier (iD) https://orcid.org/0000-0001-5326-3005
Wen-Chin Yang (iD) https://orcid.org/0000-0001-6410-2581
Yea-Lih Lin (iD) https://orcid.org/0000-0003-4063-0771
Philippe Pasero (iD) https://orcid.org/0000-0001-5891-0822

## Ethics
All mice were handled in accordance with the guidelines laid out by the Academia Sinica Institutional Animal Care and Utilization Committee (Protocol No. 10-12-097).

Reviewer #1 (Public review): https://doi.org/10.7554/eLife.104718.2.sa1
Reviewer #2 (Public review): https://doi.org/10.7554/eLife.104718.2.sa2
Author response https://doi.org/10.7554/eLife.104718.2.sa3

# Additional files

## Supplementary files
MDAR checklist

## Data availability
All data generated or analyzed during this study are included in the manuscript and supporting files; source data files have been provided for Figures 2–5.

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
