## [Editor Report · eLife Assessment]

This study presents **important** findings on the activity of two compounds, BKC and IBC, isolated from *Psoralea corylifolia*, which act synergistically to inhibit cancer cell proliferation. Using a spectrum of methods, the authors characterized the mechanisms of action of both drugs, providing **convincing** evidence that BKC targets DNA polymerases and IBC selectively inhibits CHK2. The study opens the possibility of improving the effectiveness of the combination of BKC and other damaging agents with IBC in cancer treatment.

[Editors' note: this paper was reviewed by Review Commons.]

---

## [Referee Report · Reviewer #1 (Public review)]

The manuscript by Coquel et al. investigates the effects of BKC and IBC, two compounds found in Psoralea corylifolia in DNA replication and the response to DNA damage, and explores their potential use in cancer treatment. These compounds have been previously shown to affect different cellular pathways and the authors use transformed cancer cells of different origins and a non-transformed cell line to question if their combination is toxic in cancer versus non-cancer cells. They propose that BKC inhibits DNA polymerases while IBC targets CHK2. Their results show that both compounds do affect DNA replication, inducing replication stress and affecting double strand break repair. They also show that their combined use increases their toxicity in a synergistic manner.

Comments on current version:

The authors have addressed the main questions raised in the original manuscript. The new data provide stronger evidence supporting the inhibition of DNA polymerases by BKC and the effect of IBC on CHK2. In addition, the new data provides information about the potential mechanism of action of IBC in cells and xenograft models. Together, the revised manuscript has notably increased the relevance and impact of the results with stronger conclusions and better controlled experiments.

---

## [Referee Report · Reviewer #2 (Public review)]

Summary:

The manuscript: "Synergistic effect of inhibiting CHK2 and DNA replication on cancer cell growth" successfully demonstrates that the compounds BKC and IBC found in Psoralea corylifolia act synergistically to inhibit cancer cell proliferation, using a wide range of well-chosen methodologies. Moreover, the authors characterized the mechanisms of action of both drugs, which result in inhibition of cell proliferation. The use of multiple cell lines and the mice models makes the study robust and complete.

Significance:

The manuscript presents a well written study that offers new insights and contributions to the field. Although the inhibitors described have been known in science, the authors present convincingly their mode of action, which is either better characterized (for BKC) or inhibiting a different than previously suggested enzyme (for IBC). Authors also nicely pinpoint and explain the narrow window of concentrations when these two compounds act synergistically rather than additively. The analyses in multiple cell lines, mouse models and in combination with other cancer treatments, make this study of interest not only for fundamental researchers but also for translational scientists and industry.

---

## [Author Response]

**(1) General Statements**

We thank all three reviewers for their constructive comments and suggestions. We also thank reviewers #2 and #3 for considering our work to be timely and of interest to the field, not only for basic researchers, but also for translational scientists and industry. We are now providing additional results to further support our hypothesis and hope that all reviewers will find that our manuscript is now ready for publication.

**(2) Point-by-point description of the revisions**

**Reviewer #1 (Evidence, reproducibility and clarity (Required)):**
The manuscript by Coquel et al. investigates the effects of BKC and IBC, two compounds found in Psoralea corylifolia in DNA replication and the response to DNA damage, and explores their potential use in cancer treatment. These compounds have been previously shown to affect different cellular pathways and the authors use transformed cancer cells of different origins and a non-transformed cell line to question if their combination is toxic in cancer versus non-cancer cells. They propose that BKC inhibits DNA polymerases while IBC targets CHK2. Their results show that both compounds do affect DNA replication, inducing replication stress and affecting double strand break repair. They also show that their combined use increases their toxicity in a synergistic manner.However, there are some major conclusions that are still not very well supported by the data: first, the differential effect on cancer and non-transformed cells; second, the direct link of BKC to the inhibition of DNA polymerases; and third, it is unclear if CHK2 is the relevant target for IBC in this context.Regarding these points the authors should address the following issues:(1) Most of the experiments use BJ fibroblasts as a control cell line. In order to evaluate if these compounds are preferentially toxic for cancer cells, the use of more than one non-transformed cell line is necessary. In addition, BJ cells are fibroblasts while most of the cancer cell lines employed are of epithelial origin. The authors could use MCF10 and RPE cells (both of epithelial origin) as control cell lines to complement the results and better support this claim.

We have now monitored the effect of IBC and BKC on the proliferation of MCF-7, MCF-10A and RPE-1 cells using the WST-1 assay and obtained similar results as for BJ and MCF-7 cells. These results are now included in the revised manuscript as Fig. S1A and S1B.

(2) In order to explore what are the targets of BKC and IBC Cellular Thermal Shift Assays (CETSA) could be used. Either by doing an unbiased mass spectrometry analysis of proteins stabilized by these compounds or by a direct analysis of candidate proteins by western blot (a similar approach has been used for IBC to show that it inhibits SIRT2 in Ren et al., 2024 Phytotherapy Res).

We thank this Reviewer for suggesting the use of the CETSA assay. We have now performed CETSA on MCF-7 cells and found that IBC stabilizes CHK2 but not CHK1, to the same extent as the commercial CHK2 inhibitor BML-277 used here as a positive control. These results are now shown in new Fig. 4G and 4H.

(3) For BKC in vitro polymerase assays could be carried out to show the direct inhibition of the DNA polymerase delta, for instance.

We have used high-speed *Xenopus* egg extracts to replicate ssDNA in vitro (Fig. S2C). This assay differs from the in vitro replication assay using low-speed *Xenopus* egg extracts (Fig. 2H) in that it only monitors elongation by replicative DNA polymerases (Pol δ and ε) and not earlier steps such as origin licensing and activation. The combined use of both low-speed and highspeed extracts strongly supports the view that BKC inhibits replicative DNA polymerases.

To confirm this result, we have also used CETSA to monitor BKC binding to different subunits of DNA Polδ and Polε in MCF-7 cells and in *Xenopus* egg extracts (Fig. 3C-D Fig. S3). We found that BKC binds POLD1 and POLE, the catalytic subunits of Pol δ and ε respectively, but not the accessory subunit POLD3 nor PCNA. Together with our docking results and DNA fiber experiments, these data strongly support the view that BKC is a potent inhibitor of DNA Polδ and Polδ.

(4) In addition, the authors could analyze the integrity of replication forks by PCNA immunofluorescence analysis. The colocalization of PCNA and POLD or POLE subunits could also support the role of DNA polymerases as targets of BKC.

Our molecular docking results also show that BKC occupies the catalytic sites of DNA Pol δ and ε, which may not affect their subcellular localization and/or PCNA binding. Since our DNA replication assays, CETSA and DNA fiber analyses strongly support the view that BKC inhibits replicative DNA polymerases, we have not performed this additional experiment.

(5) In the case of IBC and the inhibition of CHK2, the authors should check the effect of IBC on the phosphorylation of BRCA1 on S988. The changes in CHK2 phosphorylation in Figure 3B are not convincing. The experiment should be repeated and the average of at least three experiments needs to be quantified.

We now provide evidence that IBC inhibits BRCA1 phosphorylation on S988. Western blots and quantification for three biological replicates are shown in Fig. 4C and Fig. S4H. Densitometric quantification of CHK2 phosphorylation on S516 from 3 biological replicates, along with statistical analysis, is now shown in Fig. S4G.

(6) To prove that CHK2 is the relevant target for IBC the authors could test if ATM and CHK2 knockout cells are more resistant to this compound, since it would prevent the phosphorylation of CHK2.

We have performed siRNA transfection targeting CHK2. The transfected cells died after 72 hours in culture, so we have been unable to determine whether CHK2-KD cells have increased resistance to IBC.

In addition to these experiments, I would suggest some other major improvements in the manuscript:

(1) The concentration of both compounds should be provided in molar units throughout the paper.

Thanks for pointing this out, we now use molar units throughout the paper.

(2) The authors do not clearly indicate the concentration that is employed in the different experiments, making it difficult to assess the results. For instance, Figure 2 does not include the concentration in the legend or in the text. Time and concentration need to be clearly shown for each experiment.

The experimental conditions and inhibitor concentrations are now clearly indicated for each experiment.

(3) Some experiments are only repeated once (fiber assays) or twice (cell cycle analysis by flow cytometry). These experiments need to be repeated 3 times and the proper statistical analysis performed (comparison of the medians).

Superplots with biological replicates for all DNA fiber assays are now displayed. The number of biological replicates is now indicated in the legends and appropriate statistical analyses are used.

Other minor points or suggestions:(1) Analyzing fork asymmetry would further support the direct effect of BKC on DNA polymerases.

The effect of BKC on fork asymmetry is now shown in Fig. 2F.

(2) A dose dependent analysis of BKC on the speed of DNA replication would also support this point.

Superplots of DNA fiber assays showing the effect of different concentrations of BKC on fork speed from three biological replicates are now included in Fig. 2E.

(3) Page 7: BKC reduces fork speed ...two-fold. This sentence is not very clear, it would be better to say that speed is half of the control.

This sentence was changed to “BKC reduced fork speed by a factor of two relative to untreated cells”.

(4) Figure 4G and S4D show contradictory results regarding the induction of Rad51 foci by IBC treatment. This needs to be clarified.

Figure 4G and S4D (now Fig. 5G and S5D) do not show contradictory results. In both cases, IBC treatment impaired the induction of RAD51 foci by IR or bleomycin.

(5) Page 12, Figure S5C is called for but it does not exist (probably meaning Figure S5B).

We apologize for this error, which has now been corrected.

**Reviewer #1 (Significance):**
The work by Coquel et al. aims at elucidating the use of BKC and IBC as a combined therapy to induce cell death in cancer cells by targeting DNA replication and CHK2. Both BKC and IBC have been previously shown to affect the proliferation of cancer cells. BKC has been shown to induce S phase arrest in an ATR dependent manner in MCF7 cells (Li et al., 2016 Front Pharm), while IBC induces cell death in MDA-MB-231 cells (Wu et al., 2022 Molecules). In this regard, the more interesting contribution of the manuscript is the potential identification of the targets of these compounds in cancer cells. The inhibition of CHK2 by IBC is quite compelling although it needs to be further proven. In contrast, the hypothesis that BKC inhibits DNA polymerases remains highly speculative. The results offer a limited advance in the knowledge of the mechanism of action of these two compounds. Focusing on the action of IBC on CHK2 would increase the impact of the results. In this sense a very recent report has been published showing that IBC inhibits SIRT2 (Ren et al., 2024 Phyto Res), showing that IBC can affect multiple enzymes and processes. This should be taken into account for a further analysis of its mechanism of action.In addition to the identification of the targets of BKC and IBC, the authors also focus on their combination for cancer treatment. This is based on the idea that blocking the DSB repair and inducing replication stress at the same time is an efficient approach to induce cancer cell death. This is not a new concept, since the loss of ATM sensitizes cancer cells to the inhibition of the replication stress response and several combination therapies have been put forward with the idea of generating replication stress and preventing the subsequent repair of the double strand breaks induced in these cells. Thus, the novelty here is limited, especially considering that the effect of BKC on DNA replication has already been described. Further, since its mechanism of action is unclear, it is difficult to ascribe the observed synergy to the speculated hypothesis. A deeper analysis of IBC as a CHK2 inhibitor would be more interesting, and the potential combination with other chemotherapy agents such as replication stress inhibitors, HU or DNA damaging agents. Also, the lack of a good control of non-transformed cells also reduces the relevance of the work.In its current state, the interest of the manuscript is limited. The mechanistical advance is not strong enough and is not completely supported by the data, and the use of these compounds as a combination therapy does not provide new insights in cancer treatment. In my opinion, focusing on the inhibition of CHK2 by IBC and its potential use would broaden the impact of the results beyond the mere analysis of the action of these compounds.

We thank this reviewer for his/her constructive and insightful comments. We have followed his/her advice and focused our analysis on the action of IBC on CHK2. Using CETSA, we confirmed that IBC binds CHK2 to the same extent as BML-277 inhibitor, but does not bind CHK1. We also show that IBC inhibits BRCA1 phosphorylation on S988 and CHK2 phosphorylation on S516. Together with the results presented in the initial version of the manuscript, these data support the view that CHK2 is a key IBC target. We have also applied CETSA to DNA polymerases and confirmed that BKC directly targets DNA Polδ and ε. Although it is unlikely that IBC and BKC will ever be used in combination therapies, the synergistic effect that we measured on cancer cells in vivo and in vitro indicates that IBC sensitizes cancer cells to endogenous replication stress and to exogenous sources of DNA damage, which could be used to replace BKC in combination therapies. For instance, our data indicate that IBC can be used in combination with drugs such as etoposide, doxorubicin or cyclophosphamide to potentiate their effect on drug-resistant lymphoma cell lines (DLBCL). As requested by this Reviewer, we have modified the discussion section to put more emphasis on IBC and CHK2 inhibitors and we hope that he/she will now find this revised version suitable for publication.

**Reviewer #2 (Evidence, reproducibility and clarity):**
In the manuscript by Coquel et al., the authors report their findings on the effect of 2 natural compounds from Psoralea corylofolia plant extracts on cancer cells. They show that these compounds, bakuchiol (BKC) and isobavachalcone (IBC), inhibit proliferation of cancer cells and tumor development in xenografted mice, particularly when used in combination. They further show that BKC inhibited DNA polymerases and induced replication stress, and show evidence that IBC inhibits Chk2 kinase activity and downstream double-strand break repair. Based on their findings, the authors conclude that Chk2 inhibition and DNA replication inhibition represent a potential synergistic strategy to selecting target cancer cells.Major:(1) The data showing IBC is a Chk2 inhibitor is weak and more rigorous investigation is needed to establish this compound as a Chk2 inhibitor.

As indicate in our response to Reviewer #1, we have now analyzed the binding of IBC to CHK2 using the Cellular Thermal Shift Assay (CETSA) in MCF-7 cells. Our data clearly show that IBC binds to CHK2 but not CHK1. These results are now shown in Fig. 4G and 4H.

For one, the authors mention they screened 43 cell cycle-related kinases in vitro, but only show data for 8 kinases in their kinase activity screens. Of these 8 kinases, Chk2 is the most strongly inhibited, but there are no data shown for the other 35 kinases.

Data for all the protein kinases tested in the in vitro assay are now presented in Fig. S4D and S4E.

Additionally, the purpose of the CHK2 mutants should be discussed in the text.

The CHK2(I157T) mutation is linked to an increased risk of breast and colorectal cancers. CHK2(R145W) is associated with Li-Fraumeni Syndrome. Both mutations do not affect the basal kinase activity of CHK2. This information is now indicated in the legend of Fig. S4D.

Secondly, the western blot in Fig 3B, appears to show a very modest effect of IBC on Chk2 autophosphorylation and not that different from the effect of IBC on Akt phosphorylation in Fig S3a. Yet, the authors claim that IBC inhibits Chk2 but not Akt. To strengthen these blots, a known Chk2 inhibitor, such as the one shown in Fig 4 (BML-277) should be included as a positive control for pChk2 similarly to what was shown for Akt with MK-2206.

We have now replaced the western blot in Fig. 3B (now Fig. 4B) with another biological replicate. Quantifications and statistical analyses of biological replicates are shown in Fig. S4G. Overall, we observed a 50% reduction of CHK2 auto-phosphorylation in MCF7 cells treated with IBC, and a 20% reduction in AKT phosphorylation (Fig. S4A). There was no additional reduction in AKT phosphorylation when cells were treated with IBC in combination with MK-2206, compared to cells treated with MK-2206 alone. We now include the CHK2 inhibitor BML-277 as a positive control alongside with IBC to monitor CHK2 and CHK1 auto-phosphorylation in Fig. 4B, S4G, 4D and S4I, respectively.

Western blots showing a loss of phosphorylation of additional Chk2 targets is also needed. The manuscript mentions Brca1 S988 as a Chk2 substrate important for DSB repair. Showing the effect of IBC on this phosphorylation site would strengthen the conclusions.

We now provide evidence that IBC inhibits BRCA1 phosphorylation at S988. Western blots and quantification for three biological replicates are shown in Fig. 4C and S4H.

(2) The authors claim that the combination of IBC and BKC inhibit cell growth in a synergistic manner and that the "effect is more pronounce on cancer cells than on non-cancer cells." However, only 1 non-malignant cell line was used, and it was a fibroblast line. To make this claim, the authors need to show the effect in additional non-malignant cells, preferably with epithelial cell types.

We have now monitored cell proliferation using the WST-1 assay in two additional non-malignant cell lines, namely MCF-10A and RPE-1 cells. Cells were treated with IBC/BKC and their growth was compared to that of MCF-7 cells. These experiments yielded similar results to those obtained with BJ fibroblasts. These new data are now included in the revised version as Fig. S1A and S1B.

Minor:(1) Densitometry data for all western blots should be shown with mean+/- stdev of independent western blots.

Densitometry data for all western blots with biological replicates are now shown in supplementary figures.

(2) In Figure 1B the statistical test used to analyze cell number was not stated.

The statistical test is now indicated in Fig. 1B.

(3) In Figure 2A, the DAPI image for BKC is the merged image and should be replaced with just DAPI.

This error has now been corrected.

(4) In Figure 2B, the y-axis label says "yH2AX foci (MFI)". MFI and foci are not the same thing, and for yH2AX, the signal is often not focal. MFI of yH2AX is an appropriate measurement for replication stress, it's just not appropriate to equate MFI to foci.

We apologize for this labeling error, which has now been corrected.

(5) For the 53BP1 MFI and Rad51 MFI shown in Fig 4 and Fig S4, it is more appropriate to show the number of foci/cell as these are better indicators of breaks and repair sites. MFI is influenced by expression levels of the proteins and not necessarily the break/repair.

The numbers of 53BP1 and RAD51 foci are now shown.

(6) The data in Figures 5B and 5C are very difficult to read. Perhaps color-coat the lines/symbols.

We have now colored the graph to increase its readability.

**Reviewer #2 (Significance):**
The findings reported in this manuscript are timely, of interest to the field, and are mostly wellsupported by the experimental data. However, there are a few concerns that need to be addressed.

We are grateful to Reviewer #2 for his positive assessment of our manuscript. We hope that we have adequately addressed all of his/her specific concerns and that he/she will agree with the need to put more emphasis on IBC and CHK2 inhibition as requested by Reviewer #1.

**Reviewer #3 (Evidence, reproducibility and clarity):**
The manuscript: "Synergistic effect of inhibiting CHK2 and DNA replication on cancer cell growth" successfully demonstrates that the compounds BKC and IBC found in Psoralea corylifolia act synergistically to inhibit cancer cell proliferation, using a wide range of well-chosen methodologies. Moreover, the authors characterized the mechanisms of action of both drugs, which result in inhibition of cell proliferation. The use of multiple cell lines and the mice models makes the study robust and complete. The manuscript presents a well written study that offers new insights and contributions to the field.A few suggestions to improve the study:(1) Given that both compounds BKC and IBC have already been previously described in the literature, it would be helpful for the reader to have them described better at the beginning of the study.

Thanks for pointing this out. We have now better described BKC and IBC at the beginning of the results section, as well as in the discussion. We agree that this could be helpful to readers.

(2) Addition of western blot quantifications over the number of experimental repeats is important specifically for Fig. 2C and Fig. 3C where partial effect of treatment on a signal level is reported.

The densitometry analysis of data shown in Fig. 2C and biological replicates are now shown in Fig. S2B. Quantification for Fig. 3C (now Fig. 4D) is shown in Fig. S4I.

(3) The quantification of mean intensity for 53BP1 and RAD51 foci should be exchanged with the quantification of number of foci per cell. While the quantification of gH2AX signal intensity is a correct representation of induction of this signal upon damage, foci formed by protein recruitment to DNA damage sites should be quantified by counting the number of foci, rather than signal in the whole cell/nucleus. These proteins exist before damage and are re-located in response to the damage.

Quantification of 53BP1 and RAD51 foci is now expressed as the number of foci per cell.

(4) Materials & Methods section is missing the methods for the experiment described in Fig. 1B. In summary, after addressing our few concerns, we believe the manuscript should be accepted for publication.

The WST-1 assay used for cell number quantification is included in “Reagents” in Material & Methods section.

**Reviewer #3 (Significance):**
The manuscript presents a well written study that offers new insights and contributions to the field. Although the inhibitors described have been known in science, the authors present convincingly their mode of action, which is either better characterized (for BKC) or inhibiting a different than previously suggested enzyme (for IBC). Authors also nicely pinpoint and explain the narrow window of concentrations when these two compounds act synergistically rather than additively. The analyses in multiple cell lines, mouse models and in combination with other cancer treatments, makes this study of interest not only for fundamental researchers but also for translational scientists and industry.My field of expertise: DNA replication and replication stress across model systems.

We are grateful to Reviewer #3 for his/her very positive assessment of our work and we hope that he/she will find this revised version suitable for publication.